# On the Theory of Implicit Deep Learning: Global Convergence with Implicit Layers

**Kenji Kawaguchi**
Harvard University
Cambridge, MA 02138, USA
`kkawaguchi@fas.harvard.edu`

## Abstract

A deep equilibrium model uses implicit layers, which are implicitly defined through an equilibrium point of an infinite sequence of computation. It avoids any explicit computation of the infinite sequence by finding an equilibrium point directly via root-finding and by computing gradients via implicit differentiation. In this paper, we analyze the gradient dynamics of deep equilibrium models with nonlinearity only on weight matrices and non-convex objective functions of weights for regression and classification. Despite non-convexity, convergence to global optimum at a linear rate is guaranteed without any assumption on the width of the models, allowing the width to be smaller than the output dimension and the number of data points. Moreover, we prove a relation between the gradient dynamics of the deep implicit layer and the dynamics of trust region Newton method of a shallow explicit layer. This mathematically proven relation along with our numerical observation suggests the importance of understanding implicit bias of implicit layers and an open problem on the topic. Our proofs deal with implicit layers, weight tying and nonlinearity on weights, and differ from those in the related literature.

## 1 Introduction

A feedforward deep neural network consists of a stack of $H$ layers, where $H$ is the depth of the network. The value for the depth $H$ is typically a hyperparameter and is chosen by network designers (e.g., ResNet-101 in He et al. 2016). Each layer computes some transformation of the output of the previous layer. Surprisingly, several recent studies achieved results competitive with the state-of-the-art performances by using the same transformation for each layer with *weight tying* (Dabre & Fujita, 2019; Bai et al., 2019b; Dehghani et al., 2019). In general terms, the output of the $l$-th layer with weight tying can be written by

$$z^{(l)} = h(z^{(l-1)}; x, \theta) \quad \text{for } l = 1, 2, \dots, H-1, \tag{1}$$

where $x$ is the input to the neural network, $z^{(l)}$ is the output of the $l$-th layer (with $z^{(0)} = x$), $\theta$ represents the trainable parameters that are shared among different layers (i.e., weight tying), and $z^{(l-1)} \mapsto h(z^{(l-1)}; x, \theta)$ is some continuous function that transforms $z^{(l-1)}$ given $x$ and $\theta$. With weight tying, the memory requirement does not increase as the depth $H$ increases in the forward pass. However, the efficient backward pass to compute gradients for training the network usually requires to store the values of the intermediate layers. Accordingly, the overall computational requirement typically increases as the finite depth $H$ increases even with weight tying.

Instead of using a finite depth $H$, Bai et al. (2019a) recently introduced the *deep equilibrium model* that is equivalent to running an *infinitely* deep feedforward network with weight tying. Instead of running the layer-by-layer computation in equation (1), the deep equilibrium model uses root-finding to directly compute a fixed point $z^* = \lim_{l \to \infty} z^{(l)}$, where the limit can be ensured to exist by a choice of $h$. We can train the deep equilibrium model with gradient-based optimization by analytically backpropagating through the fixed point using implicit differentiation (e.g., Griewank & Walther, 2008; Bell & Burke, 2008; Christianson, 1994). With numerical experiments, Bai et al. (2019a) showed that the deep equilibrium model can improve performance over previous state-of-the-art models while significantly reducing memory consumption.

Despite the remarkable performances of deep equilibrium models, our theoretical understanding of its properties is yet limited. Indeed, immense efforts are still underway to mathematically understand deep *linear* networks, which have finite values for the depth $H$ without weight tying (Saxe et al., 2014; Kawaguchi, 2016; Hardt & Ma, 2017; Laurent & Brecht, 2018; Arora et al., 2018; Bartlett et al., 2019; Du & Hu, 2019; Arora et al., 2019a; Zou et al., 2020b). In deep linear networks, the function $h$ at each layer is linear in $\theta$ and linear in $x$; i.e., the map $(x, \theta) \mapsto h(z^{(l-1)}; x, \theta)$ is bilinear. Despite this linearity, several key properties of deep learning are still present in deep linear networks. For example, the gradient dynamics is nonlinear and the objective function is non-convex. Accordingly, understanding gradient dynamics of deep linear networks is considered to be a valuable step towards the mathematical understanding of deep neural networks (Saxe et al., 2014; Arora et al., 2018; 2019a).

In this paper, inspired by the previous studies of deep linear networks, we initiate a theoretical study of gradient dynamics of deep equilibrium *linear* models as a step towards theoretically understanding general deep equilibrium models. As we shall see in Section 2, the function $h$ at each layer is *nonlinear* in $\theta$ for deep equilibrium linear models, whereas it is linear for deep linear networks. This additional nonlinearity is essential to enforce the existence of the fixed point $z^*$. The additional non-linearity, the infinite depth, and weight tying are the three key proprieties of deep equilibrium linear models that are absent in deep linear networks. Because of these three differences, we cannot rely on the previous proofs and results in the literature of deep linear networks. Furthermore, we analyze gradient dynamics, whereas Kawaguchi (2016); Hardt & Ma (2017); Laurent & Brecht (2018) studied the *loss landscape* of deep linear networks. We also consider a general class of loss functions for both regression and classification, whereas Saxe et al. (2014); Arora et al. (2018); Bartlett et al. (2019); Arora et al. (2019a); Zou et al. (2020b) analyzed gradient dynamics of deep linear networks in the setting of the square loss.

Accordingly, we employ different approaches in our analysis and derive qualitatively and quantitatively different results when compared with previous studies. In Section 2, we provide theoretical and numerical observations that further motivate us to study deep equilibrium linear models. In Section 3, we mathematically prove convergence of gradient dynamics to global minima and the exact relationship between the gradient dynamics of deep equilibrium linear models and that of the adaptive trust region method. Section 5 gives a review of related literature, which strengthens the main motivation of this paper along with the above discussion (in Section 1). Finally, Section 6 presents concluding remarks on our results, the limitation of this study, and future research directions.

## 2 PRELIMINARIES

We begin by defining the notation. We are given a training dataset $((x_i, y_i))_{i=1}^n$ of $n$ samples where $x_i \in \mathcal{X} \subseteq \mathbb{R}^{m_x}$ and $y_i \in \mathcal{Y} \subseteq \mathbb{R}^{m_y}$ are the $i$-th input and the $i$-th target output, respectively. We would like to learn a hypothesis (or predictor) from a parametric family $\mathcal{H} = \{f_\theta : \mathbb{R}^{m_x} \to \mathbb{R}^{m_y} \mid \theta \in \Theta\}$ by minimizing the objective function $\mathcal{L}$ (called the empirical loss) over $\theta \in \Theta$: $\mathcal{L}(\theta) = \sum_{i=1}^n \ell(f_\theta(x_i), y_i)$, where $\theta$ is the parameter vector and $\ell : \mathbb{R}^{m_y} \times \mathcal{Y} \to \mathbb{R}_{\geq 0}$ is the loss function that measures the difference between the prediction $f_\theta(x_i)$ and the target $y_i$ for each sample. For example, when the parametric family of interest is the class of linear models as $\mathcal{H} = \{x \mapsto W\phi(x) \mid W \in \mathbb{R}^{m_y \times m}\}$, the objective function $\mathcal{L}$ can be rewritten as:

$$L_0(W) = \sum_{i=1}^n \ell(W\phi(x_i), y_i), \tag{2}$$

where the feature map $\phi$ is an arbitrary fixed function that is allowed to be nonlinear and is chosen by model designers to transforms an input $x \in \mathbb{R}^{m_x}$ into the desired features $\phi(x) \in \mathbb{R}^m$. We use $\text{vec}(W) \in \mathbb{R}^{m_y m}$ to represent the standard vectorization of a matrix $W \in \mathbb{R}^{m_y \times m}$.

Instead of linear models, our interest in this paper lies on *deep equilibrium models*. The output $z^*$ of the last hidden layer of a deep equilibrium model is defined by

$$z^* = \lim_{l \to \infty} z^{(l)} = \lim_{l \to \infty} h(z^{(l-1)}; x, \theta) = h(z^*; x, \theta), \tag{3}$$

where the last equality follows from the continuity of $z \mapsto h(z; x, \theta)$ (i.e., the limit commutes with the continuous function). Thus, $z^*$ can be computed by solving the equation $z^* = h(z^*; x, \theta)$ without running the infinitely deep layer-by-layer computation. The gradients with respect to parameters are computed analytically via backpropagation through $z^*$ using implicit differentiation.

## 2.1 Deep Equilibrium Linear Models

A deep equilibrium *linear* model is an instance of the family of deep equilibrium models and is defined by setting the function $h$ at each layer as follows:

$$h(z^{(l-1)}; x, \theta) = \gamma\sigma(A)z^{(l-1)} + \phi(x), \tag{4}$$

where $\theta = (A, B)$ with two trainable parameter matrices $A \in \mathbb{R}^{m \times m}$ and $B \in \mathbb{R}^{m_y \times m}$. Along with a positive real number $\gamma \in (0, 1)$, the nonlinear function $\sigma$ is used to ensure the existence of the fixed point and is defined by $\sigma(A)_{ij} = \frac{\exp(A_{ij})}{\sum_{k=1}^{m} \exp(A_{kj})}$. The class of deep equilibrium linear models is given by $\mathcal{H} = \{x \mapsto B\left(\lim_{l\to\infty} z^{(l)}(x, A)\right) \mid A \in \mathbb{R}^{m \times m}, B \in \mathbb{R}^{m_y \times m}\}$, where $z^{(l)}(x, A) = \gamma\sigma(A)z^{(l-1)} + \phi(x)$. Therefore, the objective function for deep equilibrium linear models can be written as

$$L(A, B) = \sum_{i=1}^{n} \ell\left(B\left(\lim_{l\to\infty} z^{(l)}(x_i, A)\right), y_i\right). \tag{5}$$

The outputs of deep equilibrium linear models $f_\theta(x) = B\left(\lim_{l\to\infty} z^{(l)}(x, A)\right)$ are nonlinear and non-multilinear in the optimization variable $A$. This is in contrast to linear models and deep linear networks. From the optimization viewpoint, linear models $W\phi(x)$ are called linear because they are linear in the optimization variables $W$. Deep linear networks $W^{(H)}W^{(H-1)} \cdots W^{(1)}x$ are multi-linear in the optimization variables $(W^{(1)}, W^{(2)}, \ldots, W^{(H)})$ (this holds also when we replace $x$ by $\phi(x)$). This difference creates a challenge in the analysis of deep equilibrium linear models.

Following previous works on gradient dynamics of different machine learning models (Saxe et al., 2014; Ji & Telgarsky, 2020), we consider the process of learning deep equilibrium linear models via gradient flow:

$$\frac{d}{dt}A_t = -\frac{\partial L}{\partial A}(A_t, B_t), \quad \frac{d}{dt}B_t = -\frac{\partial L}{\partial B}(A_t, B_t), \quad \forall t \geq 0, \tag{6}$$

where $(A_t, B_t)$ represents the model parameters at time $t$ with an arbitrary initialization $(A_0, B_0)$. Throughout this paper, a feature map $\phi$ and a real number $\gamma \in (0, 1)$ are given and arbitrary (except in experimental observations) and we omit their universal quantifiers for the purpose of brevity.

## 2.2 Preliminary Observation for Additional Motivation

Our analysis is chiefly motivated as a step towards mathematically understanding general deep equilibrium models (as discussed in Sections 1 and 5). In addition to the main motivation, this section provides supplementary motivations through theoretical and numerical preliminary observations.

In general deep equilibrium models, the limit, $\lim_{l\to\infty} z^{(l)}$, is not ensured to exist (see Appendix C). In this view, the class of deep equilibrium linear models is one instance where the limit is guaranteed to exist for any values of model parameters as stated in Proposition 1:

**Proposition 1.** *Given any $(x, A)$, the sequence $(z^{(l)}(x, A))_l$ in Euclidean space $\mathbb{R}^m$ converges.*

*Proof.* We use the nonlinearity $\sigma$ to ensure the convergence in our proof in Appendix A.5. □

Proposition 1 shows that we can indeed define the deep equilibrium linear model with $\lim_{l\to\infty} z^{(l)} = z^*(x, A)$. Therefore, understanding this model is a sensible starting point for theory of general deep equilibrium models.

As our analysis has been mainly motivated for theory, it would be of additional value to discuss whether the model would also make sense in practice, at least potentially in the future. Consider an (unknown) underling data distribution $P(x, y) = P(y|x)P(x)$. Intuitively, if the mean of the $P(y|x)$ is approximately given by a (true unknown) deep equilibrium linear model, then it would make sense to use the parametric family of deep equilibrium linear models to have the inductive bias in practice. To confirm this intuition, we conducted numerical simulations. To generate datasets, we first drew uniformly at random 200 input images for input data points $x_i$ from a standard image dataset — CIFAR-10, CIFAR-100 or Kuzushiji-MNIST (Krizhevsky & Hinton, 2009; Clanuwat et al., 2019). We then generated targets as $y_i = B^*(\lim_{l\to\infty} z^{(l)}(x_i, A^*)) + \delta_i$ where $\delta_i \overset{\text{i.i.d.}}{\sim} \mathcal{N}(0, 1)$. Each entry of the true (unknown) matrices $A^*$ and $B^*$ was independently drawn from the standard normal distribution. For each dataset generated in this way, we used stochastic gradient descent (SGD) to train linear models, fully-connected feedforward deep neural networks with ReLU nonlinearity

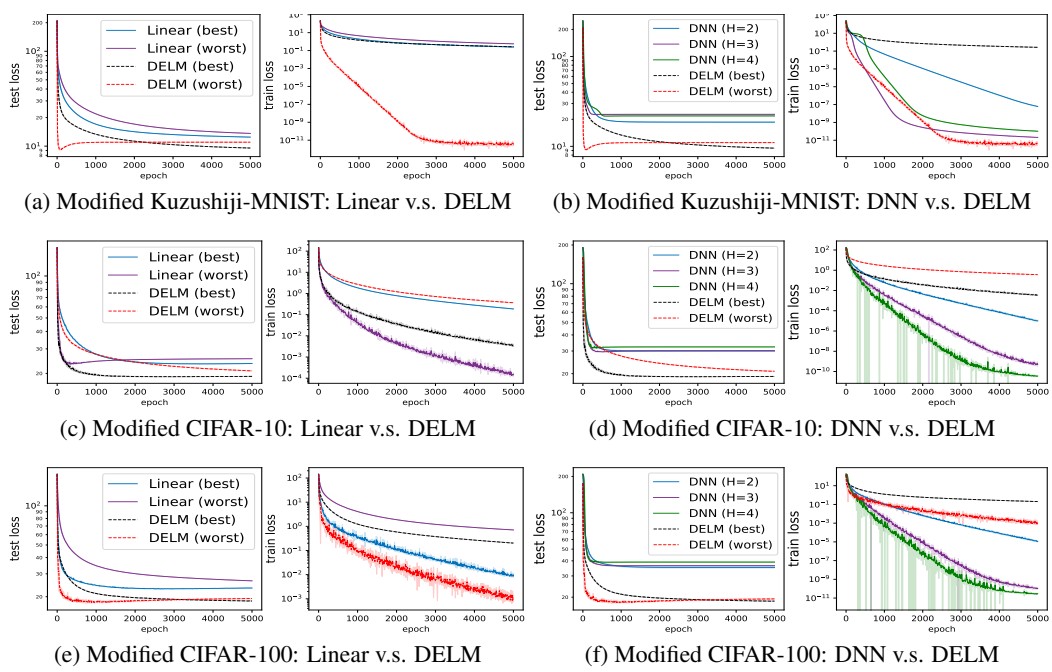

(a) Modified Kuzushiji-MNIST: Linear v.s. DELM     (b) Modified Kuzushiji-MNIST: DNN v.s. DELM

(c) Modified CIFAR-10: Linear v.s. DELM     (d) Modified CIFAR-10: DNN v.s. DELM

(e) Modified CIFAR-100: Linear v.s. DELM     (f) Modified CIFAR-100: DNN v.s. DELM

Figure 1: Preliminary observations for additional motivation to theoretically understand deep equilibrium linear models. The figure shows test and train losses versus the number of epochs for linear models, deep equilibrium linear models (DELMs), and deep neural networks with ReLU (DNNs).

(DNNs), and deep equilibrium linear models. For all models, we fixed $\phi(x) = x$. See Appendix D for more details of the experimental settings.

The results of this numerical test are presented in Figure 1. In the figure, the plotted lines indicate the mean values over five random trials whereas the shaded regions represent error bars with one standard deviation. The plots for linear models and deep equilibrium linear models are shown with the best and worst learning rates (separately for each model in terms of the final test errors at epoch = 5000) from the set of learning rates $S_{\text{LR}} = \{0.01, 0.005, 0.001, 0.0005, 0.0001, 0.00005\}$. The plots for DNNs are shown with the best learning rates (separately for each depth $H$) from the set $S_{\text{LR}}$. As can be seen, all models preformed approximately the same at initial points, but deep equilibrium linear models outperformed both linear models and DNNs in test errors after training, confirming our intuition above. Moreover, we confirmed qualitatively same behaviors with four more datasets as well as for DNNs with and without bias terms in Appendix D. These observations additionally motivated us to study deep equilibrium linear models to obtain our main results in the next section. The purpose of these experiments is to provide a secondary motivation for our theoretical analyses.

## 3 MAIN RESULTS

In this section, we establish mathematical properties of gradient dynamics for deep equilibrium linear models by directly analyzing its trajectories. We prove linear convergence to global minimum in Section 3.1 and further analyze the dynamics from the viewpoint of trust region in Section 3.2.

### 3.1 CONVERGENCE ANALYSIS

We begin in Section 3.1.1 with a presentation of the concept of the Polyak-Łojasiewicz (PL) inequality and additional notation. The PL inequality is used to regularize the choice of the loss functions $\ell$ in our main convergence theorem for a general class of losses in Section 3.1.2. We conclude in Section 3.1.3 by providing concrete examples of the convergence theorem with the square loss and the logistic loss, where the PL inequality is no longer required as the PL inequality is proven to be satisfied by these loss functions.

### 3.1.1 THE POLYAK-ŁOJASIEWICZ INEQUALITY AND ADDITIONAL NOTATION

In our context, the notion of the PL inequality is formally defined as follows:

**Definition 1.** The function $L_0$ is said to satisfy *the Polyak-Łojasiewicz (PL) inequality with radius $R \in (0, \infty]$ and parameter $\kappa > 0$* if $\frac{1}{2}\|\nabla L_0^{\text{vec}}(\text{vec}(W))\|_2^2 \geq \kappa(L_0^{\text{vec}}(\text{vec}(W)) - L_{0,R}^*)$ for all $\|W\|_1 < R$, where $L_0^{\text{vec}}(\text{vec}(\cdot)) := L_0(\cdot)$ and $L_{0,R}^* := \inf_{W:\|W\|_1 < R} L_0(W)$.

With any radius $R > 0$ sufficiently large (such that it covers the domain of $L_0$), Definition 1 becomes equivalent to the definition of the PL inequality in the optimization literature (e.g., Polyak, 1963; Karimi et al., 2016). See Appendix C for additional explanations on the equivalence. In general, the non-convex objective function $L$ of deep equilibrium linear models does not satisfy the PL inequality. Therefore, we cannot assume the inequality on $L$. However, in order to obtain linear convergence for a general class of the loss functions $\ell$, we need some assumption on $\ell$: otherwise, we can choose a loss $\ell$ to violate the convergence. Accordingly, we will regularize the choice of the loss $\ell$ through the PL inequality on the function $L_0 : W \mapsto \sum_{i=1}^n \ell(W\phi(x_i), y_i)$.

The PL inequality with a radius $R \in (0, \infty]$ (Definition 1) leads to the notion of the global minimum value in the domain corresponding to the radius in our analysis: $L_R^* = \inf_{A \in \mathbb{R}^{m \times m}, B \in \mathcal{B}_R} L(A, B)$, where $\mathcal{B}_R = \{B \in \mathbb{R}^{m_y \times m} \mid \|B\|_1 < (1-\gamma)R\}$. With $R = \infty$, this recovers the global minimum value $L^*$ in the unconstrained domain as $L_R^* = L^* := \inf_{A \in \mathbb{R}^{m \times m}, B \in \mathbb{R}^{m_y \times m}} L(A, B)$. Furthermore, if a global minimum $(A^*, B^*) \in \mathbb{R}^{m \times m} \times \mathbb{R}^{m_y \times m}$ exists, there exists $\bar{R} < \infty$ such that for any $R \in [\bar{R}, \infty)$, we have $B^* \in \mathcal{B}_R$ and thus $L_R^* = L^*$. In other words, if a global minimum exists, using a (sufficiently large) finite radius $R < \infty$ suffices to obtain $L_R^* = L^*$.

We close this subsection by introducing additional notation. For a real symmetric matrix $M$, we use $\lambda_{\min}(M)$ to represent its smallest eigenvalue. For an arbitrary matrix $M \in \mathbb{R}^{d \times d'}$, we let $\text{rank}(M)$ be its rank, $\|M\|_p$ be its matrix norm induced by the vector $p$-norm, $\sigma_{\min}(M)$ be its smallest singular value (i.e., the $\min(d, d')$-th largest singular value), $M_{*j}$ be its $j$-th column vector in $\mathbb{R}^d$, and $M_{i*}$ be its $i$-th row vector in $\mathbb{R}^{d'}$. For $d \in \mathbb{N}_{>0}$, we denote by $I_d$ the identify matrix in $\mathbb{R}^{d \times d}$. We define the Jacobian matrix $J_{k,t} \in \mathbb{R}^{m \times m}$ of the vector-valued function $A_{*k} \mapsto \sigma(A)_{*k}$ by $(J_{k,t})_{ij} = \frac{\partial \sigma(A)_{ik}}{\partial A_{jk}}\big|_{A=A_t}$ for all $t \geq 0$ and $k = 1, \ldots, m$. Finally, we define the feature matrix $\Phi \in \mathbb{R}^{m \times n}$ by $\Phi_{ki} = \phi(x_i)_k$ for $k = 1, \ldots, m$ and $i = 1, \ldots, n$ .

### 3.1.2 Main Convergence Theorem

Using the PL inequality only on the loss function $\ell$ through $L_0$ (Definition 1), we present our main theorem — a guarantee on linear convergence to global minimum for the gradient dynamics of the non-convex objective $L$ for deep equilibrium linear models:

**Theorem 1.** *Let $\ell : \mathbb{R}^{m_y} \times \mathcal{Y} \to \mathbb{R}_{\geq 0}$ be arbitrary such that the function $q \mapsto \ell(q, y_i)$ is differentiable for any $i \in \{1, \ldots, n\}$ (with an arbitrary $m_y \in \mathbb{N}_{>0}$ and an arbitrary $\mathcal{Y}$). Then, for any $T > 0$, $R \in (0, \infty]$ and $\kappa > 0$ such that $\|B_t\|_1 < (1-\gamma)R$ for all $t \in [0, T]$ and $L_0$ satisfies the PL inequality with the radius $R$ and the parameter $\kappa$, the following holds:*

$$L(A_T, B_T) \leq L_R^* + \left(L(A_0, B_0) - L_{0,R}^*\right) e^{-2\kappa\lambda_T T}, \tag{7}$$

*where $\lambda_T := \inf_{t \in [0,T]} \lambda_{\min}(D_t) > 0$ and $D_t$ is a positive definite matrix defined by*

$$D_t := \sum_{k=1}^m \left[(U_t^{-\top})_{*k}(U_t^{-1})_{k*} \otimes \left(I_{m_y} + \gamma^2 B_t U_t^{-1} J_{k,t} J_{k,t}^\top U_t^{-\top} B_t^\top\right)\right], \tag{8}$$

*with $U_t := I_m - \gamma\sigma(A_t)$. Furthermore, $\lambda_T \geq \frac{1}{m(1+\gamma)^2}$ for any $T \geq 0$ ($\lim_{T \to \infty} \lambda_T \geq \frac{1}{m(1+\gamma)^2}$).*

*Proof.* The additional nonlinearity $\sigma$ creates a complex interaction among $m$ hidden neurons. This interaction is difficult to be factorized out for the gradients of $L$ with respect to $A$. This is different from but analogous to the challenge to deal with nonlinear activations in the loss landscape of (non-overparameterized) deep nonlinear networks, for which previous works have made assumptions of sparse connections to factorize the interaction (Kawaguchi et al., 2019). In contrast, we do not rely on sparse connections. Instead, we observe that although it is difficult to factorize this complex interaction (due to the nonlinearity $\sigma$) in the space of loss landscape, we can factorize it in the space of gradient dynamics. See Appendix A.1 for the proof overview and the complete proof. $\square$

Theorem 1 shows that in the worst case for $\lambda_T$, the optimality gap decreases exponentially towards zero as $L(A_T, B_T) - L_R^* \leq C_0 e^{-\frac{2\kappa}{m(1+\gamma)^2}T}$, where $C_0 = L(A_0, B_0) - L_{0,R}^*$. Therefore, for any

desired accuracy $\epsilon > 0$, setting $C_0 e^{-\frac{2\kappa}{m(1+\gamma)^2}T} \leq \epsilon$ and solving for $T$ yield that

$$L(A_T, B_T) - L_R^* \leq \epsilon \quad \text{for any } T \geq \frac{m(1+\gamma)^2}{2\kappa} \log \frac{L(A_0, B_0) - L_{0,R}^*}{\epsilon}. \tag{9}$$

Theorem 1 also states that the rate of convergence improves further depending on the quality of the matrix $D_t$ (defined in equation (8)) in terms of its smallest eigenvalue over the particular trajectory $(A_t, B_t)$ up to the specific time $t \leq T$; i.e., $\lambda_T = \inf_{t \in [0,T]} \lambda_{\min}(D_t)$. This opens up the direction of future work for further improvement of the convergence rate through the design of initialization $(A_0, B_0)$ to maximize $\lambda_T$ for trajectories generated from a specific initialization scheme.

### 3.1.3 EXAMPLES: SQUARE LOSS AND LOGISTIC LOSS

The main convergence theorem in the previous subsection is stated for any radius $R \in (0, \infty]$ and parameter $\kappa > 0$ that satisfy the conditions on $\|B_t\|_1$ and the PL inequality (see Theorem 1). The values of these variables are not completely specified there as they depend on the choice of the loss functions $\ell$. In this subsection, we show that these values can be specified further and the condition on PL inequality can be discarded by considering a specific choice of loss functions $\ell$.

In particular, by using the square loss for $\ell$, we prove that we can set $R = \infty$ and $\kappa = 2\sigma_{\min}(\Phi)^2$:

**Corollary 1.** Let $\ell(q, y_i) = \|q - y_i\|_2^2$ where $y_i \in \mathbb{R}^{m_y}$ for $i = 1, 2, \ldots, n$ (with an arbitrary $m_y \in \mathbb{N}_{>0}$). Assume that $\mathrm{rank}(\Phi) = \min(n, m)$. Then for any $T > 0$,

$$L(A_T, B_T) \leq L^* + (L(A_0, B_0) - L_0^*) e^{-4\sigma_{\min}(\Phi)^2 \lambda_T T},$$

where $\sigma_{\min}(\Phi) > 0$, $L_0^* := \inf_{W \in \mathbb{R}^{m_y \times m}} L_0(W)$, and $\lambda_T := \inf_{t \in [0,T]} \lambda_{\min}(D_t) \geq \frac{1}{m(1+\gamma)^2}$.

*Proof.* This statement follows from Theorem 1. The conditions on $\|B_t\|_1$ and the PL inequality (in Theorem 1) are now discarded by using the property of the square loss $\ell$. See Appendix A.3 for the complete proof. □

In Corollary 1, the global linear convergence is established for the square loss without the notion of the radius $R$ as we set $R = \infty$. Even with the square loss, the objective function $L$ is non-convex. Despite the non-convexity, Corollary 1 shows that for any desired accuracy $\epsilon > 0$,

$$L(A_T, B_T) - L^* \leq \epsilon \quad \text{for any } T \geq \frac{m(1+\gamma)^2}{4\sigma_{\min}(\Phi)^2} \log \frac{L(A_0, B_0) - L_0^*}{\epsilon}. \tag{10}$$

Corollary 1 allows both cases of $m \leq n$ and $m > n$. In the case of over-parameterization $m > n$, the covariance matrix $\Phi\Phi^\top \in \mathbb{R}^{m \times m}$ (or $XX^\top$ with $\phi(x) = x$) is always rank deficient because $\mathrm{rank}(\Phi\Phi^\top) = \mathrm{rank}(\Phi) \leq n < m$. This implies that the Hessian of $L_0$ is always rank deficient, because the Hessian of $L_0$ is $\nabla^2 L_0^{\mathrm{vec}}(\mathrm{vec}(W)) = 2[\Phi\Phi^\top \otimes I_{m_y}] \in \mathbb{R}^{m_y m \times m_y m}$ (see Appendix A.3 for its derivation) and because $\mathrm{rank}([\Phi\Phi^\top \otimes I_{m_y}]) = \mathrm{rank}(\Phi\Phi^\top)\mathrm{rank}(I_{m_y}) \leq m_y n < m_y m$. Since the strong convexity on a twice differentiable function requires its Hessian to be of full rank, this means that the objective $L_0$ for linear models is not strongly convex in the case of over-parameterization $m > n$. Nevertheless, we establish the linear convergence to global minimum for deep equilibrium linear models in Corollary 1 for both cases of $m > n$ and $m \leq n$ by using Theorem 1.

For the logistic loss for $\ell$, the following corollary proves the global convergence at a linear rate:

**Corollary 2.** Let $\ell(q, y_i) = -y_i \log(\frac{1}{1+e^{-q}}) - (1 - y_i) \log(1 - \frac{1}{1+e^{-q}}) + \tau \|q\|_2^2$ with an arbitrary $\tau \geq 0$ where $y_i \in \{0, 1\}$ for $i = 1, 2, \ldots, n$. Assume that $\mathrm{rank}(\Phi) = m$. Then for any $T > 0$ and $R \in (0, \infty]$ such that $\|B_t\|_1 < (1 - \gamma)R$ for all $t \in [0, T]$, the following holds:

$$L(A_T, B_T) \leq L_R^* + (L(A_0, B_0) - L_{0,R}^*) e^{-2(2\tau + \rho(R))\sigma_{\min}(\Phi)^2 \lambda_T T},$$

where $\sigma_{\min}(\Phi) > 0$, $\lambda_T := \inf_{t \in [0,T]} \lambda_{\min}(D_t) \geq \frac{1}{m(1+\gamma)^2}$, and

$$\rho(R) := \inf_{\substack{W : \|W\|_1 < R, \\ i \in \{1, \ldots, n\}}} \left(\frac{1}{1 + e^{-W\phi(x_i)}}\right)\left(1 - \frac{1}{1 + e^{-W\phi(x_i)}}\right) \geq 0.$$

*Proof.* This statement follows from Theorem 1 by proving that the condition on PL inequality is satisfied with the parameter $\kappa = (2\tau + \rho(R))\sigma_{\min}(\Phi)^2$. See Appendix A.4 for the complete proof. □

In Corollary 2, we can also set $R = \infty$ to remove the notion of the radius $R$ from the statement of the global convergence for the logistic loss. By setting $R = \infty$, Corollary 2 states that for any $T > 0$,

$$L(A_T, B_T) \leq L^* + (L(A_0, B_0) - L_0^*) e^{-4\tau\sigma_{\min}(\Phi)^2\lambda_T T},$$

for the logistic loss. For any $\tau > 0$, this implies that for any desired accuracy $\epsilon > 0$,

$$L(A_T, B_T) - L^* \leq \epsilon \quad \text{for any } T \geq \frac{m(1+\gamma)^2}{4\tau\sigma_{\min}(\Phi)^2} \log \frac{L(A_0, B_0) - L_0^*}{\epsilon}. \tag{11}$$

In practice, we may want to set $\tau > 0$ to regularize the parameters (for generalization) and to ensure the existence of global minima (for optimization and identifiability). That is, if we set $\tau = 0$ instead, the global minima may not exist in any bounded space, due to the property of the logistic loss. This is consistent with Corollary 2 in that if $\tau = 0$, equation (11) does not hold and we must consider the convergence to the global minimum value $L_R^*$ defined in a bounded domain with a radius $R < \infty$. In the case of $\tau = 0$ and $R < \infty$, Corollary 2 implies that for desired accuracy $\epsilon > 0$,

$$L(A_T, B_T) - L_R^* \leq \epsilon \quad \text{for any } T \geq \frac{m(1+\gamma)^2}{2\rho(R)\sigma_{\min}(\Phi)^2} \log \frac{L(A_0, B_0) - L_{0,R}^*}{\epsilon}, \tag{12}$$

where we have $\rho(R) > 0$ because $R < \infty$. Therefore, Corollary 2 establish the linear convergence to global minimum with both cases of $\tau > 0$ and $\tau = 0$ for the logistic loss.

### 3.2 Understanding Dynamics Through Trust Region Newton Method

In this subsection, we analyze the dynamics of deep equilibrium linear models in the space of the hypothesis, $f_{\theta_t} : x \mapsto B_t \left( \lim_{l \to \infty} z^{(l)}(x, A_t) \right)$. For any functions $g$ and $\bar{g}$ with a domain $\mathcal{X} \subseteq \mathbb{R}^{m_x}$, we write $g = \bar{g}$ if $g(x) = \bar{g}(x)$ for all $x \in \mathcal{X}$.

The following theorem shows that the dynamics of deep equilibrium linear models $f_{\theta_t}$ can be written as $\frac{d}{dt} f_{\theta_t} = \frac{1}{\delta_t} V_t \phi$ where $\frac{1}{\delta_t}$ is scalar and $V_t$ follows the dynamics of a trust region Newton method of shallow models with the (non-standard) adaptive trust region $\mathcal{V}_t$. This suggests potential benefits of deep equilibrium linear models in two aspects: when compared to shallow models, it can sometimes accelerate optimization via the effect of the implicit trust region method (but not necessarily as the trust region method does not necessarily accelerate optimization) and induces novel implicit bias for generalization via the non-standard implicit trust region $\mathcal{V}_t$.

**Theorem 2.** *Let $\ell : \mathbb{R}^{m_y} \times \mathcal{Y} \to \mathbb{R}_{\geq 0}$ be arbitrary such that the function $q \mapsto \ell(q, y_i)$ is differentiable for any $i \in \{1, \ldots, n\}$ with $m_y = 1$ and (an arbitrary $\mathcal{Y}$). Then for any time $t \geq 0$, there exist a real number $\bar{\delta}_t > 0$ such that for any $\delta_t \in (0, \bar{\delta}_t]$,*

$$\frac{d}{dt} f_{\theta_t} = \frac{1}{\delta_t} V_t \phi, \quad \text{vec}(V_t) \in \underset{v \in \mathcal{V}_t}{\arg\min} \, L_0^t(v), \tag{13}$$

*where $\mathcal{V}_t := \{v \in \mathbb{R}^m : \|v\|_{G_t} \leq \delta_t \|\frac{d}{dt} \text{vec}(B_t U_t^{-1})\|_{G_t}\}$, $G_t := U_t \left(S_t^{-1} - \delta_t F_t\right) U_t^\top \succ 0$, and*

$$L_0^t(v) := L_0^{\text{vec}}(\text{vec}(B_t U_t^{-1})) + \nabla L_0^{\text{vec}}(\text{vec}(B_t U_t^{-1}))^\top v + \frac{1}{2} v^\top \nabla^2 L_0^{\text{vec}}(\text{vec}(B_t U_t^{-1})) v.$$

*Here, $F_t := \sum_{i=1}^n \nabla^2 \ell_i(f_{\theta_t}(x_i))(\lim_{l \to \infty} z^{(l)}(x_i, A_t))(\lim_{l \to \infty} z^{(l)}(x_i, A_t))^\top$ with $\ell_i(q) := \ell(q, y_i)$ and $S_t := I_m + \gamma^2 \text{diag}(v_t^S)$ with $v_t^S \in \mathbb{R}^m$ and $(v_t^S)_k := \|J_{k,t}^\top (B_t U_t^{-1})^\top\|_2^2 \; \forall k$.*

*Proof.* This is proven with the Karush–Kuhn–Tucker (KKT) conditions for the constrained optimization problem: $\text{minimize}_{v \in \mathcal{V}_t} L_0^t(v)$. See Appendix A.2. $\square$

When many global minima exist, a difference in the gradient dynamics can lead to a significant discrepancy in the learned models: i.e., two different gradient dynamics can find significantly different global minima with different behaviors for generalization and test accuracies (Kawaguchi et al., 2017). In machine learning, this is an important phenomenon called *implicit bias* — inductive bias induced implicitly through gradient dynamics — and is the subject of an emerging active research area (Gunasekar et al., 2017; Soudry et al., 2018; Gunasekar et al., 2018; Woodworth et al., 2020; Moroshko et al., 2020).

As can be seen in Theorem 2, the gradient dynamics of deep equilibrium linear models $f_{\theta_t}$ differs from that of linear models $W_t\phi$ with any adaptive learning rates, fixed preconditioners, and existing variants of Newton methods. This is consistent with our experiments in Section 2.2 and Appendix D where the dynamics of deep equilibrium linear models resulted in the learned predictors with higher test accuracies, when compared to linear models with any learning rates. In this regard, Theorem 2 provides a partial explanation (and a starting point of the theory) for the observed generalization behaviors, whereas Theorem 1 (with Corollaries 1 and 2) provides the theory for the global convergence observed in the experiments.

Theorem 2, along with our experimental results, suggests the importance of theoretically understanding implicit bias of the dynamics with the time-dependent trust region. In Appendix B, we show that Theorem 2 suggests a new type of implicit bias towards a simple function as a result of infinite depth, whereas understanding this implicit bias in more details is left as an open problem for future work.

## 4 EXPERIMENTS

In this section, we conduct experiments to further verify and demonstrate our theory. To compare with the previous findings, we use the same synthetic data as that in the previous work (Zou et al., 2020b): i.e., we randomly generate $x_i \in \mathbb{R}^{10}$ from the standard normal distribution and set $y_i = -x_i + 0.1\varsigma_i$ for all $i \in \{1, 2, \ldots, n\}$ with $n = 1000$, where $\varsigma_i$ is independently generated by the standard normal distribution. We set $\phi(x) = x$ and use the square loss $\ell(q, y_i) = \|q - y_i\|_2^2$. As in the previous work, we consider random initialization and identity initialization (Zou et al., 2020b) and report the results in Figure 2 (a). As can be seen in the figure, deep equilibrium linear models converges to the global minimum value with all initialization and random trials, whereas linear ResNet converges to a suboptimal value with identity initialization. This is consistent with our theory for deep equilibrium linear models and the previous work for ResNet (Zou et al., 2020b).

We repeated the same experiment by generating $(x_i)_k$ independently from the uniform distribution of the interval $[-1, 1]$ instead for all $i \in \{1, \ldots, n\}$ and $k \in \{1, \ldots, m\}$ with $n = 1000$ and $m = 10$. Figure 2 (b) shows the results of this experiment with the uniform distribution and confirm the global convergence of deep equilibrium linear models again with all initialization and random trials. In this case, linear ResNet with identity initialization also converged to the global minimum value. These observations are consistent with Corollary 1 where deep equilibrium linear models are guaranteed to converge to the global minimum value without any condition on the initialization.

We now consider the rate of the global convergence. In Corollary 1, we can set $\lambda_T = \frac{1}{m(1+\gamma)^2}$ to get a guarantee for the global linear convergence rate for *all* initializations in theory. However, in practice, this is a pessimistic convergence rate and we may want to choose $\lambda_T$ depending on a initialization. To demonstrate this, using the same data as that in Figure 2 (a), Figure 2 (c) reports the numerical training trajectory along with theoretical upper bounds with initialization-independent $\lambda_T = \frac{1}{m(1+\gamma)^2}$ and initialization-dependent $\lambda_T = \inf_{t \in [0,T]} \lambda_{\min}(D_t)$. As can be seen in Figure 2 (c), the theoretical bound with initialization-dependent $\lambda_T$ demonstrates a faster and more accurate convergence rate. A qualitatively same observation is reported for the logistic loss in Appendix D.2.

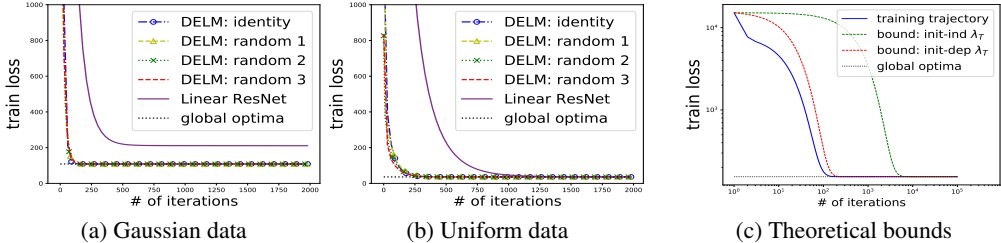

| (a) Gaussian data | (b) Uniform data | (c) Theoretical bounds |

Figure 2: (a)-(b): Convergence performances for deep equilibrium linear models (DELMs) with identity initialization and random initialization of three random trials, and linear ResNet with identity initialization. (c) the numerical training trajectory of DELMs with random initialization along with theoretical upper bounds with initialization-independent $\lambda_T$ and initialization-dependent $\lambda_T$.

## 5  RELATED WORK

The theoretical study of gradient dynamics of deep networks with some linearized component is a highly active area of research. Recently, Bartlett et al. (2019); Du & Hu (2019); Arora et al. (2019a); Zou et al. (2020b) analyzed gradient dynamics of deep linear networks and proved global convergence rates for the square loss under certain assumptions on the dataset, initialization, and network structures. For example, the dataset is assumed to be *whitened* (i.e., $\Phi\Phi^\top = I_m$ or $XX^\top = I_{m_x}$) and the initial loss is assumed to be smaller than the loss of any rank-deficient solution by Arora et al. (2019a): the input and output layers are assumed to represent special transformations and are fixed during training by Zou et al. (2020b).

Deep networks are also linearized implicitly in the neural tangent kernel (NTK) regime with significant over-parameterization $m \gg n$ (Yehudai & Shamir, 2019; Lee et al., 2019). By significantly increasing model parameters (or more concretely the width $m$), we can ensure deep features or corresponding NTK to stay nearly the same during training. In other words, deep networks in this regime are approximately linear models with random features corresponding to the NTK at random initialization. Because of this implicit linearization, deep networks in the NTK regime are shown to achieve globally minimum training errors by interpolating all training data points (Zou et al., 2020a; Li & Liang, 2018; Jacot et al., 2018; Du et al., 2019; 2018; Chizat et al., 2019; Arora et al., 2019b; Allen-Zhu et al., 2019; Lee et al., 2019; Fang et al., 2020; Montanari & Zhong, 2020).

These previous studies have significantly advanced our theoretical understanding of deep learning through the study of deep linear networks and implicitly linearized deep networks in the NTK regime. In this context, this paper is expected to contribute to the theoretical advancement through the study of a new and significantly different type of deep models — deep equilibrium linear models. In deep equilibrium linear models, the function at each layer $A \mapsto h(z^{(l-1)}; x, \theta)$ is nonlinear due to the additional nonlinearity $\sigma$: $A \mapsto h(z^{(l-1)}; x, \theta) := \gamma\sigma(A)z^{(l-1)} + \phi(x)$. In contrast, for deep linear networks, the function at each layer $W^{(l)} \mapsto h^{(l)}(z^{(l-1)}; x, W^{(l)}) := W^{(l)}z^{(l-1)}$ is linear (it is linear also with skip connection). Furthermore, the nonlinearity $\sigma$ is not an element-wise function, which poses an additional challenge in the mathematical analysis of deep equilibrium linear models. The nonlinearity $\sigma$, the infinite depth, and weight tying in deep equilibrium linear models necessitated us to develop new approaches in our proofs. The differences in the models and proofs naturally led to qualitatively and quantitatively different results. For example, we do not require any of over-parameterization $m \gg n$, interpolation of all training data points, and any assumptions mentioned above for deep linear networks.

Unlike previous papers, we also related the dynamics of deep equilibrium linear models to that of a trust region Newton method of shallow models with $G_t$-quadratic norm. This suggested potential benefits of deep equilibrium linear models. Our theory is consistent with our numerical observations.

## 6  CONCLUSION

For deep equilibrium linear models, despite the non-convexity, we have rigorously proven convergence of gradient dynamics to global minima, at a linear rate, for a general class of loss functions, including the square loss and logistic loss. Moreover, we have proven the relationship between the gradient dynamics of deep equilibrium linear models and that of the adaptive trust region method. These results apply to models with any configuration on the width of hidden layers, the number of data points, and input/output dimensions, allowing rank-deficient covariance matrices as well as both under-parameterization and over-parameterization.

The crucial assumption for our analysis is the differentiability of the function $q \mapsto \ell(q, y_i)$, which is satisfied by standard loss functions, such as the square loss, the logistic loss, and the smoothed hinge loss $\ell(q, y_i) = (\max\{0, 1 - y_iq\})^k$ with $k \geq 2$. However, it is not satisfied by the (non-smoothed) hinge loss $\ell(q, y_i) = \max\{0, 1 - y_iq\}$, the treatment of which is left to future work. Future work also includes corresponding theoretical analyses with stochastic gradient descent.

Our theoretical results (in Section 3) and numerical observations (in Section 2.2 and Appendix D) uncover the special properties of deep equilibrium linear models, providing a basis of future work for theoretical studies of implicit bias and for further empirical investigations of deep equilibrium models. In our proofs, the treatments of the additional nonlinearity $\sigma$, the infinite depth, and weight tying are especially unique, and we expect our new proof techniques to be proven useful in further studies of gradient dynamics for deep models.

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

## A  PROOFS

In this appendix, we complete the proofs of our theoretical results. We present the proofs of Theorem 1 in Appendix A.1, Theorem 2 in Appendix A.2, Corollary 1 in Appendix A.3, Corollary 2 in Appendix A.4, and Proposition 1 in Appendix A.5. We also provide a proof overview of Theorem 1 in the beginning of Appendix A.1.

Before starting our proofs, we first introduce additional notation used in the proofs and then discuss alternative proofs using the implicit function theorem to avoid relying on the convergence of Neumann series.

**Additional notation.**  Given a scalar-valued function $a \in \mathbb{R}$ and a matrix $M \in \mathbb{R}^{d \times d'}$, we write

$$\frac{\partial a}{\partial M} = \begin{bmatrix} \frac{\partial a}{\partial M_{11}} & \cdots & \frac{\partial a}{\partial M_{1d'}} \\ \vdots & \ddots & \vdots \\ \frac{\partial a}{\partial M_{d1}} & \cdots & \frac{\partial a}{\partial M_{dd'}} \end{bmatrix} \in \mathbb{R}^{d \times d'},$$

where $M_{ij}$ represents the $(i, j)$-th entry of the matrix $M$. Given a vector-valued function $a \in \mathbb{R}^d$ and a column vector $b \in \mathbb{R}^{d'}$, we write

$$\frac{\partial a}{\partial b} = \begin{bmatrix} \frac{\partial a_1}{\partial b_1} & \cdots & \frac{\partial a_1}{\partial b_{d'}} \\ \vdots & \ddots & \vdots \\ \frac{\partial a_d}{\partial b_1} & \cdots & \frac{\partial a_d}{\partial b_{d'}} \end{bmatrix} \in \mathbb{R}^{d \times d'},$$

where $b_i$ represents the $i$-th entry of the column vector $b$. Similarly, given a vector-valued function $a \in \mathbb{R}^d$ and a row vector $b \in \mathbb{R}^{1 \times d'}$, we write

$$\frac{\partial a}{\partial b} = \begin{bmatrix} \frac{\partial a_1}{\partial b_{11}} & \cdots & \frac{\partial a_1}{\partial b_{1d'}} \\ \vdots & \ddots & \vdots \\ \frac{\partial a_d}{\partial b_{11}} & \cdots & \frac{\partial a_d}{\partial b_{1d'}} \end{bmatrix} \in \mathbb{R}^{d \times d'},$$

where $b_{1i}$ represents the $i$-th entry of the row vector $b$. We use $\nabla_A L$ to represent the map $(A, B) \mapsto \frac{\partial L}{\partial A}(A, B)$ (without the usual transpose used in vector calculus). Given a matrix $M$ and a function $\varphi$, we define $\nabla_M \varphi$ similarly as the map $M \mapsto \frac{\partial \varphi}{\partial M}(M)$. Our proofs also use the indicator function:

$$\mathbb{1}\{i = k\} = \begin{cases} 1 & \text{if } i = k \\ 0 & \text{if } i \neq k \end{cases}$$

Finally, we recall the definition of the Kronecker product of two matrices: for matrices $M \in \mathbb{R}^{d_M \times d'_M}$ and $\bar{M} \in \mathbb{R}^{d_{\bar{M}} \times d'_{\bar{M}}}$,

$$M \otimes \bar{M} = \begin{bmatrix} M_{11} \bar{M} & \cdots & M_{1d'_M} \bar{M} \\ \vdots & \ddots & \vdots \\ M_{d_M 1} \bar{M} & \cdots & M_{d_M d'_M} \bar{M} \end{bmatrix} \in \mathbb{R}^{d_M d_{\bar{M}} \times d'_M d'_{\bar{M}}}.$$

**On alternative proofs using the implicit function theorem.**  In our default proofs, we utilize the Neumann series $\sum_{k=0}^{\infty} \gamma^k \sigma(A)^k$ when deriving the formula of the gradients with respect to $A$. Instead of using the Neumann series, we can alternatively use the implicit function theorem to derive the formula of the gradients with respect to $A$. Specifically, in this alternative proof, we apply the implicit function theorem to the function $\psi$ defined by

$$\psi(\text{vec}[A], z) = z - \gamma \sigma(A) z - \phi(x),$$

where $\text{vec}[A]$ and $z \in \mathbb{R}^m$ are independent variables of the function $\psi$: i.e., $\psi(\text{vec}[A], z)$ is allowed to be nonzero. On the other hand, the vector $z$ satisfying $\psi(\text{vec}[A], z) = 0$ is the fixed point

$z^* = \lim_{l \to \infty} z^{(l)}$ based on equation (3). Therefore, by applying the implicit function theorem to the function $\psi$, it holds that if the the Jacobian matrix $\frac{\partial \psi(\text{vec}[A], z)}{\partial z}\big|_{z=z^*}$ is invertible, then

$$\frac{\partial z^*}{\partial \text{vec}[A]} = -\left(\frac{\partial \psi(\text{vec}[A], z)}{\partial z}\bigg|_{z=z^*}\right)^{-1} \left(\frac{\partial \psi(\text{vec}[A], z)}{\partial \text{vec}[A]}\bigg|_{z=z^*}\right). \tag{14}$$

Since $\frac{\partial \psi(\text{vec}[A], z)}{\partial z}\big|_{z=z^*} = I - \gamma\sigma(A)$ is invertible, it holds that

$$\frac{\partial z^*}{\partial \text{vec}[A]} = -\left(I - \gamma\sigma(A)\right)^{-1} \left(\frac{\partial \psi(\text{vec}[A], z)}{\partial \text{vec}[A]}\bigg|_{z=z^*}\right). \tag{15}$$

Moreover, since $\sigma(A)z \in \mathbb{R}^m$ is a column vector,

$$
\begin{aligned}
\frac{\partial \psi(\text{vec}[A], z)}{\partial \text{vec}[A]}\bigg|_{z=z^*} &= -\gamma \frac{\partial \sigma(A)z}{\partial \text{vec}[A]}\bigg|_{z=z^*} = -\gamma \frac{\partial \text{vec}[\sigma(A)z]}{\partial \text{vec}[A]}\bigg|_{z=z^*} \\
&= -\gamma \frac{\partial [z^\top \otimes I_m]\text{vec}[\sigma(A)]}{\partial \text{vec}[A]}\bigg|_{z=z^*} \\
&= -\gamma[(z^*)^\top \otimes I_m]\frac{\partial \text{vec}[\sigma(A)]}{\partial \text{vec}[A]}.
\end{aligned} \tag{16}
$$

Combining equations (15) and (16), we have

$$\frac{\partial z^*}{\partial \text{vec}[A]} = \gamma\left(I - \gamma\sigma(A)\right)^{-1}[(z^*)^\top \otimes I_m]\frac{\partial \text{vec}[\sigma(A)]}{\partial \text{vec}[A]}. \tag{17}$$

In our proofs, whenever we require the gradients with respect to $A$, we can directly use equation (17), instead of relying on the convergence of the Neumann series. For example, equation (21) in the proof of Theorem 1 is identical to equation (17) with additional multiplication of $B_{q*}$: i.e., for the left hand side,

$$B_{q*}\frac{\partial z^*}{\partial \text{vec}[A]} = \frac{\partial B_{q*}z^*}{\partial \text{vec}[A]} = \frac{\partial B_{q*}U^{-1}\phi(x)}{\partial A}$$

and for the right hand side,

$$\gamma B_{q*}\left(I - \gamma\sigma(A)\right)^{-1}[(z^*)^\top \otimes I_m]\frac{\partial \text{vec}[\sigma(A)]}{\partial \text{vec}[A]}$$

$$= \gamma\left[\left(\frac{\partial \sigma(A)_{*1}}{\partial A_{*1}}\right)^\top (B_{q*}U^{-1})^\top \phi(x)^\top (U^{-\top})_{*1} \quad \cdots \quad \left(\frac{\partial \sigma(A)_{*m}}{\partial A_{*m}}\right)^\top (B_{q*}U^{-1})^\top \phi(x)^\top (U^{-\top})_{*m}\right]$$

## A.1 PROOF OF THEOREM 1

We begin with a proof overview of Theorem 1. We first compute the derivatives of the output of deep equilibrium linear models with respect to the parameters $A$ in Appendix A.1.1. Then using the derivatives, we rearrange the formula of $\nabla_A L$ such that it is related to the formula of $\nabla L_0$ in Appendices A.1.1–A.1.3. Intuitively, we then want to understand $\nabla_A L$ through the property of $\nabla L_0$, similarly to the landscape analyses of deep linear networks by Kawaguchi (2016). However, we note there that the additional nonlinearity $\sigma$ creates a complex interaction over the dimension $m$ to prevent us from using such a proof approach. Instead, using the proven relation of $\nabla_A L$ and $\nabla L_0$ from Appendices A.1.1–A.1.3, we directly analyze the trajectories of the dynamics over time $t$ in Appendices A.1.4–A.1.5, which results in a partial factorization of the iteration over the dimension $m$. Using such a partial factorization, we derive the linear convergence rate in Appendices A.1.6–A.1.7 by using the PL inequality and the properties of induced norms.

Before getting into the details of the proof, we now briefly discuss the property of our proof in terms of the tightness of a bound. In the condition of $\|B_t\|_1 < (1 - \gamma)R$ in the statement of Theorem 1, the quantity $(1 - \gamma)$ comes from the proof in Appendix A.1.7: i.e., it is the reciprocal of the quantity $\frac{1}{1-\gamma}$ in the upper bound of $\|(I_m - \gamma\sigma(A))^{-1}\|_1 \leq \frac{1}{1-\gamma}$. Therefore, a natural question is whether or not we can improve this bound further. This bound turns out to be tight based on the following lower bound. The matrix $I_m - \gamma\sigma(A)$ is a $Z$-matrix since off-diagonal entries are less than or equal to

zero. Furthermore, $I_m - \gamma\sigma(A)$ is $M$-matrix since eigenvalues of $I_m - \gamma\sigma(A)$ are the eigenvalues of $I_m - \gamma\sigma(A)^\top$ and the eigenvalues of $I - \gamma\sigma(A)^\top$ are lower bounded by $1 - \gamma > 0$. This is because $\sigma(A)^\top$ is a stochastic matrix with the largest eigenvalue being one. Moreover, in the proof in Appendix A.1.7, we showed that $|I - \gamma\sigma(A)|_{jj} - \sum_{i\neq j}|I - \gamma\sigma(A)|_{ij} = 1 - \gamma$ for all $j$. Therefore, using the lower bound by Morača (2008), we have

$$\|(I - \gamma\sigma(A))^{-1}\|_1 \geq \frac{1}{\max_j(|I - \gamma\sigma(A)|_{jj} - \sum_{i\neq j}|I - \gamma\sigma(A)|_{ij})} = \frac{1}{1 - \gamma},$$

which matches with the upper bound of $\|(I_m - \gamma\sigma(A))^{-1}\|_1 \leq \frac{1}{1-\gamma}$. Therefore, we cannot further improve the our bound on $\|B_t\|_1$ in general without making some additional assumption.

### A.1.1 REARRANGING THE FORMULA OF $\nabla_A L$

We will use the following facts for matrix calculus (that can be derived by using definition of derivatives: e.g., see Barnes, 2006):

$$\frac{\partial M^{-1}}{\partial a} = -M^{-1}\frac{\partial M}{\partial a}M^{-1}$$

$$\frac{\partial a^\top M^{-1} b}{\partial M} = -M^{-\top}ab^\top M^{-\top}$$

$$\frac{\partial g(M)}{\partial a} = \sum_i \sum_j \frac{\partial g(M)}{\partial M_{ij}}\frac{\partial M_{ij}}{\partial a}$$

$$\frac{\partial g(a)}{\partial M} = \frac{\partial g(a)}{\partial a}\frac{\partial a}{\partial M}$$

Recall that $U = I - \gamma\sigma(A)$. From the above facts, given a function $g$, we have

$$\begin{aligned}
\frac{\partial g(U)}{\partial A_{kl}} &= \sum_{i=1}^m \sum_{j=1}^m \frac{\partial g(U)}{\partial U_{ij}}\frac{\partial U_{ij}}{\partial A_{kl}} \\
&= \sum_{i=1}^m \sum_{j=1}^m \frac{\partial g(U)}{\partial U_{ij}}\frac{\partial U_{ij}}{\partial \sigma(A)_{ij}}\frac{\partial \sigma(A)_{ij}}{\partial A_{kl}} \\
&= -\gamma \sum_{i=1}^m \sum_{j=1}^m \frac{\partial g(U)}{\partial U_{ij}}\frac{\partial \sigma(A)_{ij}}{\partial A_{kl}}.
\end{aligned} \tag{18}$$

Using the quotient rule,

$$\begin{aligned}
\frac{\partial \sigma(A)_{ij}}{\partial A_{kl}} &= \frac{\partial}{\partial A_{kl}}\frac{\exp(A_{ij})}{\sum_t \exp(A_{tj})} \\
&= \frac{(\frac{\partial \exp(A_{ij})}{\partial A_{kl}})(\sum_t \exp(A_{tj})) - \exp(A_{ij})(\frac{\partial \sum_t \exp(A_{tj})}{\partial A_{kl}})}{(\sum_t \exp(A_{tj}))^2} \\
&= \frac{\mathbb{1}\{i = k\}\mathbb{1}\{j = l\}\exp(A_{ij})(\sum_t \exp(A_{tj})) - \mathbb{1}\{j = l\}\exp(A_{ij})\exp(A_{kj})}{(\sum_t \exp(A_{tj}))^2} \\
&= \frac{\mathbb{1}\{i = k\}\mathbb{1}\{j = l\}\exp(A_{ij})}{\sum_t \exp(A_{tj})} - \frac{\mathbb{1}\{j = l\}\exp(A_{ij})\exp(A_{kj})}{(\sum_t \exp(A_{tj}))^2} \\
&= \mathbb{1}\{i = k\}\mathbb{1}\{j = l\}\sigma(A)_{ij} - \mathbb{1}\{j = l\}\frac{\exp(A_{ij})}{\sum_t \exp(A_{tj})}\frac{\exp(A_{kj})}{\sum_t \exp(A_{tj})} \\
&= \mathbb{1}\{j = l\}\mathbb{1}\{i = k\}\sigma(A)_{ij} - \mathbb{1}\{j = l\}\sigma(A)_{ij}\sigma(A)_{kj}.
\end{aligned} \tag{19}$$

Thus,

$$\frac{\partial g(U)}{\partial A_{kl}} = -\gamma \sum_{i=1}^m \frac{\partial g(U)}{\partial U_{il}}\frac{\partial \sigma(A)_{il}}{\partial A_{kl}} = -\gamma \left(\frac{\partial g(U)}{\partial U_{*l}}\right)^\top \frac{\partial \sigma(A)_{*l}}{\partial A_{kl}} \in \mathbb{R},$$

where $\frac{\partial g(U)}{\partial U_{*l}} \in \mathbb{R}^{m \times 1}$ and $\frac{\partial \sigma(A)_{*l}}{\partial A_{kl}} \in \mathbb{R}^{m \times 1}$. This yields

$$\frac{\partial g(U)}{\partial A_{*l}} = -\gamma \left( \frac{\partial g(U)}{\partial U_{*l}} \right)^{\top} \frac{\partial \sigma(A)_{*l}}{\partial A_{*l}} \in \mathbb{R}^{1 \times m},$$

where $\frac{\partial \sigma(A)_{*l}}{\partial A_{*l}} \in \mathbb{R}^{m \times m}$.

Now we want to set $g(U)$ to be the output of deep equilibrium linear models as $g(U) = B_{q*} \left( \lim_{l \to \infty} z^{(l)}(x, A) \right)$ for each $q \in \{1, \dots, m_y\}$. To do this, we first simplify the formula of the output $B_{q*} \left( \lim_{l \to \infty} z^{(l)}(x, A) \right)$ using the following:

$$(I_m - \gamma\sigma(A)) \left( \sum_{k=0}^{l} \gamma^k \sigma(A)^k \right)$$
$$= I_m - \gamma\sigma(A) + \gamma\sigma(A) - (\gamma\sigma(A))^2 + (\gamma\sigma(A))^2 - (\gamma\sigma(A))^3 + \cdots - (\gamma\sigma(A))^{l+1}$$
$$= I - (\gamma\sigma(A))^{l+1}.$$

Therefore,

$$(I_m - \gamma\sigma(A)) \left( \lim_{l \to \infty} z^{(l)}(x, A) \right) = \lim_{l \to \infty} (I_m - \gamma\sigma(A)) \left( \sum_{k=0}^{l} \gamma^k \sigma(A)^k \phi(x) \right)$$
$$= \left( I_m - \lim_{l \to \infty} (\gamma\sigma(A))^{l+1} \right) \phi(x)$$
$$= \phi(x)$$

where the first line, the second line and the last line used the fact that $\gamma\sigma(A)_{ij} \geq 0$,

$$\|\sigma(A)\|_1 = \max_j \sum_i |\sigma(A)_{ij}| = 1,$$

and hence $\|\gamma\sigma(A)\|_1 < 1$ for $\gamma \in (0, 1)$. This shows that $B \left( \lim_{l \to \infty} z^{(l)}(x, A) \right) = BU^{-1}\phi(x)$, where the inverse $U^{-1}$ exists as the corresponding Neumann series converges $\sum_{k=0}^{\infty} \gamma^k \sigma(A)^k$ since $\|\gamma\sigma(A)\|_1 < 1$. Therefore, we can now set $g(U) = B_{q*} \left( \lim_{l \to \infty} z^{(l)}(x, A) \right) = B_{q*}U^{-1}\phi(x)$.

Then, using $\frac{\partial a^{\top} M^{-1} b}{\partial M} = -M^{-\top}ab^{\top}M^{-\top}$,

$$\frac{\partial g(U)}{\partial U} = \frac{\partial B_{q*}U^{-1}\phi(x)}{\partial U} = -U^{-\top}(B_{q*})^{\top}\phi(x)^{\top}U^{-\top},$$

which implies that

$$\frac{\partial B_{q*}U^{-1}\phi(x)}{\partial U_{*l}} = -(U^{-\top}(B_{q*})^{\top}\phi(x)^{\top}U^{-\top})_{*l}$$
$$= -U^{-\top}(B_{q*})^{\top}\phi(x)^{\top}(U^{-\top})_{*l} \in \mathbb{R}^{m \times 1}. \tag{20}$$

Combining (18) and (20),

$$\frac{\partial g(U)}{\partial A_{*l}} = \frac{\partial B_{q*}U^{-1}\phi(x)}{\partial A_{*l}} = -\gamma \left( \frac{\partial g(U)}{\partial U_{*l}} \right)^{\top} \frac{\partial \sigma(A)_{*l}}{\partial A_{*l}}$$
$$= \gamma \left( U^{-\top}(B_{q*})^{\top}\phi(x)^{\top}(U^{-\top})_{*l} \right)^{\top} \left( \frac{\partial \sigma(A)_{*l}}{\partial A_{*l}} \right)$$
$$= \gamma((U^{-\top})_{*l})^{\top}\phi(x)B_{q*}U^{-1} \left( \frac{\partial \sigma(A)_{*l}}{\partial A_{*l}} \right)$$

$$= \gamma (U^{-1})_{l*} \phi(x) B_{q*} U^{-1} \left( \frac{\partial \sigma(A)_{*l}}{\partial A_{*l}} \right) \in \mathbb{R}^{1 \times m},$$

where we used $(U^{-\top})_{*l} = ((U^{-1})^{\top})_{*l} = ((U^{-1})_{l*})^{\top}$ and $((U^{-\top})_{*l})^{\top} = (((U^{-1})_{l*})^{\top})^{\top} = (U^{-1})_{l*}$. By taking transpose,

$$\left( \frac{\partial B_{q*} U^{-1} \phi(x)}{\partial A} \right)_{*l} = \gamma \left( \frac{\partial \sigma(A)_{*l}}{\partial A_{*l}} \right)^{\top} (B_{q*} U^{-1})^{\top} \phi(x)^{\top} (U^{-\top})_{*l} \in \mathbb{R}^{m \times 1}.$$

By rearranging this to the matrix form,

$$\frac{\partial B_{q*} U^{-1} \phi(x)}{\partial A} \tag{21}$$

$$= \gamma \left[ \left( \frac{\partial \sigma(A)_{*1}}{\partial A_{*1}} \right)^{\top} (B_{q*} U^{-1})^{\top} \phi(x)^{\top} (U^{-\top})_{*1} \quad \cdots \quad \left( \frac{\partial \sigma(A)_{*m}}{\partial A_{*m}} \right)^{\top} (B_{q*} U^{-1})^{\top} \phi(x)^{\top} (U^{-\top})_{*m} \right],$$

where $\frac{\partial B_{q*} U^{-1} \phi(x)}{\partial A} \in \mathbb{R}^{m \times m}$. Each entry of this matrix represents the derivatives of the model output with respect to the parameters $A$. We now use this to rearrange $\nabla_A L(A, B) := \frac{\partial L(A,B)}{\partial A}$. We set $\hat{y}_{iq} = B_{q*} U^{-1} \phi(x)$ and $\hat{y}_i = B U^{-1} \phi(x)$ and define

$$J_k := \frac{\partial \sigma(A)_{*k}}{\partial A_{*k}} \in \mathbb{R}^{m \times m},$$

and

$$Q := \sum_{i=1}^{n} \left( \frac{\partial \ell(\hat{y}_i, y_i)}{\partial \hat{y}_i} \right)^{\top} \phi(x_i)^{\top} \in \mathbb{R}^{m_y \times m}.$$

Then, using the chain rule and the above formula of $\frac{\partial B_{q*} U^{-1} \phi(x)}{\partial A}$,

$$\frac{\partial L(A, B)}{\partial A}$$

$$= \sum_{i=1}^{n} \sum_{q=1}^{m_y} \frac{\partial \ell(\hat{y}_i, y_i)}{\partial \hat{y}_{iq}} \frac{\partial \hat{y}_{iq}}{\partial A}$$

$$= \gamma \sum_{i=1}^{n} \sum_{q=1}^{m_y} \frac{\partial \ell(\hat{y}_i, y_i)}{\partial \hat{y}_{iq}} \left[ J_1^{\top} (B_{q*} U^{-1})^{\top} \phi(x_i)^{\top} (U^{-\top})_{*1} \quad \cdots \quad J_m^{\top} (B_{q*} U^{-1})^{\top} \phi(x_i)^{\top} (U^{-\top})_{*m} \right]$$

$$= \gamma \sum_{i=1}^{n} \left[ J_1^{\top} (\sum_{q=1}^{m_y} (B_{q*} U^{-1})^{\top} \frac{\partial \ell(\hat{y}_i, y_i)}{\partial \hat{y}_{iq}}) \phi(x_i)^{\top} (U^{-\top})_{*1} \quad \cdots \quad J_m^{\top} (\sum_{q=1}^{m_y} (B_{q*} U^{-1})^{\top} \frac{\partial \ell(\hat{y}_i, y_i)}{\partial \hat{y}_{iq}}) \phi(x_i)^{\top} (U^{-\top})_{*m} \right]$$

$$= \gamma \sum_{i=1}^{n} \left[ J_1^{\top} (\sum_{q=1}^{m_y} ((B U^{-1})^{\top})_{*q} \frac{\partial \ell(\hat{y}_i, y_i)}{\partial \hat{y}_{iq}}) \phi(x_i)^{\top} (U^{-\top})_{*1} \quad \cdots \quad J_m^{\top} (\sum_{q=1}^{m_y} ((B U^{-1})^{\top})_{*q} \frac{\partial \ell(\hat{y}_i, y_i)}{\partial \hat{y}_{iq}}) \phi(x_i)^{\top} (U^{-\top})_{*m} \right]$$

$$= \gamma \sum_{i=1}^{n} \left[ J_1^{\top} (B U^{-1})^{\top} (\frac{\partial \ell(\hat{y}_i, y_i)}{\partial \hat{y}_i})^{\top} \phi(x_i)^{\top} (U^{-\top})_{*1} \quad \cdots \quad J_m^{\top} (B U^{-1})^{\top} (\frac{\partial \ell(\hat{y}_i, y_i)}{\partial \hat{y}_i})^{\top} \phi(x_i)^{\top} (U^{-\top})_{*m} \right]$$

$$= \gamma \left[ \left( \frac{\partial \sigma(A)_{*1}}{\partial A_{*1}} \right)^{\top} (B U^{-1})^{\top} Q (U^{-\top})_{*1} \quad \cdots \quad \left( \frac{\partial \sigma(A)_{*m}}{\partial A_{*m}} \right)^{\top} (B U^{-1})^{\top} Q (U^{-\top})_{*m} \right]$$

Summarizing the above, we have that

$$\nabla_A L(A, B) := \frac{\partial L(A, B)}{\partial A} = \gamma \left[ J_1^{\top} (B U^{-1})^{\top} Q (U^{-\top})_{*1} \quad \cdots \quad J_m^{\top} (B U^{-1})^{\top} Q (U^{-\top})_{*m} \right], \tag{22}$$

where $\nabla_A L(A, B) \in \mathbb{R}^{m \times m}$.

### A.1.2 REARRANGING THE FORMULA OF $\nabla L_0$

In order to relate $L_0$ to the gradient dynamics of $L$, we now rearrange the formula of $\nabla L_0$. We set $\hat{y}_{iq} = W_{q*}\phi(x_i) \in \mathbb{R}$ and $\hat{y}_i = W\phi(x_i) \in \mathbb{R}^{m_y}$ for linear models. Then, by the chain rule,

$$\frac{\partial L_0(W)}{\partial W} = \sum_{i=1}^{n}\sum_{q=1}^{m_y} \frac{\partial \ell(\hat{y}_i, y_i)}{\partial \hat{y}_{iq}} \frac{\partial \hat{y}_{iq}}{\partial W}.$$

Since

$$\frac{\partial \hat{y}_{iq}}{\partial W_{k*}} = \mathbb{1}\{k = q\}\phi(x_i)^\top,$$

we have

$$\frac{\partial L_0(W)}{\partial W_{k*}} = \sum_{i=1}^{n} \frac{\partial \ell(\hat{y}_i, y_i)}{\partial \hat{y}_{ik}} \frac{\partial \hat{y}_{ik}}{\partial W_{k*}} = \sum_{i=1}^{n} \frac{\partial \ell(\hat{y}_i, y_i)}{\partial \hat{y}_{ik}}\phi(x_i)^\top.$$

By rearranging this into the matrix form,

$$\frac{\partial L_0(W)}{\partial W} = \begin{bmatrix} \sum_{i=1}^{n} \frac{\partial \ell(\hat{y}_i,y_i)}{\partial \hat{y}_{i1}}\phi(x_i)^\top \\ \vdots \\ \sum_{i=1}^{n} \frac{\partial \ell(\hat{y}_i,y_i)}{\partial \hat{y}_{im_y}}\phi(x_i)^\top \end{bmatrix}$$

$$= \sum_{i=1}^{n} \begin{bmatrix} \frac{\partial \ell(\hat{y}_i,y_i)}{\partial \hat{y}_{i1}}\phi(x_i)^\top \\ \vdots \\ \frac{\partial \ell(\hat{y}_i,y_i)}{\partial \hat{y}_{im_y}}\phi(x_i)^\top \end{bmatrix}$$

$$= \sum_{i=1}^{n} \begin{bmatrix} \frac{\partial \ell(\hat{y}_i,y_i)}{\partial \hat{y}_{i1}} \\ \vdots \\ \frac{\partial \ell(\hat{y}_i,y_i)}{\partial \hat{y}_{im_y}} \end{bmatrix} \phi(x_i)^\top$$

$$= \sum_{i=1}^{n} \left(\frac{\partial \ell(\hat{y}_i,y_i)}{\partial \hat{y}_i}\right)^\top \phi(x_i)^\top \in \mathbb{R}^{m_y \times m}$$

where $\frac{\partial \ell(\hat{y}_i,y_i)}{\partial \hat{y}_i} \in \mathbb{R}^{1 \times m_y}$. Thus,

$$\nabla L_0(W) := \frac{\partial L(W)}{\partial W} = \sum_{i=1}^{n} \left(\frac{\partial \ell(\hat{y}_i, y_i)}{\partial \hat{y}_i}\right)^\top \phi(x_i)^\top \in \mathbb{R}^{m_y \times m}. \tag{23}$$

### A.1.3 COMBINING THE FORMULA OF $\nabla_A L$ AND $\nabla L_0$

Combining (22) and (23) by resolving the different definitions of $\hat{y}_i$ yields that

$$\nabla_A L(A, B) \tag{24}$$
$$= \gamma \left[ J_1^\top (BU^{-1})^\top \nabla L_0(BU^{-1})(U^{-\top})_{*1} \quad \cdots \quad J_m^\top (BU^{-1})^\top \nabla L_0(BU^{-1})(U^{-\top})_{*m} \right].$$

Here, if there is no additional nonlinearity $\sigma$, the matrices $J_k = \frac{\partial \sigma(A)_{*k}}{\partial A_{*k}}$ become identity for all $k$. In that case, $\nabla_A L(A, B)$ can be further simplified and factorize over $m$, which is desired for the analysis of gradient dynamics. However, due to the additional nonlinearity, we cannot factorize $\nabla_A L(A, B)$ over $m$. One of the key techniques in our analysis is to keep this un-factorized $\nabla_A L(A, B)$ and find a way to factorize it during the update of parameters $(A_t, B_t)$ in the gradient dynamics, as shown later in this proof. To do so, we now start considering the dynamics over time $t$.

### A.1.4 ANALYSING $\left(\lim_{l\to\infty} z^{(l)}(x, A_t)\right)$

Now let us temporarily consider a gradient dynamics discretized by the Euler method as

$$A_{t+1} = A_t - \alpha \nabla_A L(A_t, B_t),$$

with some step size $\alpha > 0$. Then,

$$\left(\lim_{l\to\infty} z^{(l)}(x, A_{t+1})\right) = (I_m - \gamma\sigma(A_{t+1}))^{-1}\phi(x)$$
$$= (I_m - \gamma\sigma(A_t - \alpha\nabla_A L(A_t, B_t)))^{-1}\phi(x),$$

where we used $\left(\lim_{l\to\infty} z^{(l)}(x, A_t)\right) = U_t^{-1}\phi(x)$ from Section A.1.1. By setting $\varphi_{ij}(\alpha) = \sigma(A - \alpha\nabla_A L(A, B))_{ij} \in \mathbb{R}$,

$$\sigma(A - \alpha\nabla_A L(A, B))_{ij} = \varphi_{ij}(\alpha) = \varphi_{ij}(0) + \frac{\partial\varphi_{ij}(0)}{\partial\alpha}\alpha + O(\alpha^2).$$

By using the chain rule and setting $M = A - \alpha\nabla_A L(A, B) \in \mathbb{R}^{n\times n}$,

$$\frac{\partial\varphi_{ij}(\alpha)}{\partial\alpha} = \sum_{k=1}^{m}\sum_{l=1}^{m}\frac{\partial\sigma(M)_{ij}}{\partial M_{kl}}\frac{\partial M_{kl}}{\partial\alpha}$$
$$= -\sum_{k=1}^{m}\sum_{l=1}^{m}[\mathbb{1}\{j = l\}\mathbb{1}\{i = k\}\sigma(M)_{ij} - \mathbb{1}\{j = l\}\sigma(M)_{ij}\sigma(M)_{kj}]\nabla_A L(A, B)_{kl}$$
$$= -\sum_{k=1}^{m}[\mathbb{1}\{i = k\}\sigma(M)_{ij} - \sigma(M)_{ij}\sigma(M)_{kj}]\nabla_A L(A, B)_{kj}$$

Therefore,

$$\frac{\partial\varphi_{ij}(0)}{\partial\alpha} = -\sum_{k=1}^{m}[\mathbb{1}\{i = k\}\sigma(A)_{ij} - \sigma(A)_{ij}\sigma(A)_{kj}]\nabla_A L(A, B)_{kj}$$
$$= -\sum_{k=1}^{m}\frac{\partial\sigma(A)_{ij}}{\partial A_{kj}}\nabla_A L(A, B)_{kj}$$
$$= -\frac{\partial\sigma(A)_{ij}}{\partial A_{*j}}\nabla_A L(A, B)_{*j} \in \mathbb{R}.$$

Recalling the definition of $J_k := \frac{\partial\sigma(A)_{*k}}{\partial A_{*k}} \in \mathbb{R}^{m\times m}$,

$$\frac{\partial\varphi_{*j}(0)}{\partial\alpha} = -\frac{\partial\sigma(A)_{*j}}{\partial A_{*j}}\nabla_A L(A, B)_{*j} = -J_j\nabla_A L(A, B)_{*j} \in \mathbb{R}^{m\times 1}$$

Rearranging it into the matrix form,

$$\frac{\partial\varphi(0)}{\partial\alpha} = -[J_1\nabla_A L(A, B)_{*1} \quad \cdots \quad J_m\nabla_A L(A, B)_{*m}] \in \mathbb{R}^{m\times m}$$

Putting the above equations together,

$$\sigma(A - \alpha\nabla_A L(A, B)) = \varphi(0) + \alpha\frac{\partial\varphi(0)}{\partial\alpha} + O(\alpha^2)$$
$$= \sigma(A) - \alpha[J_1\nabla_A L(A, B)_{*1} \quad \cdots \quad J_m\nabla_A L(A, B)_{*m}] + O(\alpha^2).$$

Thus,

$$[I_m - \gamma\sigma(A - \alpha\nabla_A L(A, B))]^{-1}$$
$$= [I_m - \gamma[\sigma(A) - \alpha[J_1\nabla_A L(A, B)_{*1} \quad \cdots \quad J_m\nabla_A L(A, B)_{*m}] + O(\alpha^2)]]^{-1}$$
$$= [I_m - \gamma\sigma(A) + \gamma\alpha[J_1\nabla_A L(A, B)_{*1} \quad \cdots \quad J_m\nabla_A L(A, B)_{*m}] + O(\alpha^2)]^{-1}$$
$$= [U + \alpha\gamma[J_1\nabla_A L(A, B)_{*1} \quad \cdots \quad J_m\nabla_A L(A, B)_{*m}] + O(\alpha^2)]^{-1}.$$

By setting $M = [J_1\nabla_A L(A, B)_{*1} \quad \cdots \quad J_m\nabla_A L(A, B)_{*m}]$ and $\varphi(\alpha) = [U + \alpha\gamma M + o(\alpha^2)]^{-1}$ and by using $\frac{\partial\bar{M}^{-1}}{\partial a} = -\bar{M}^{-1}\frac{\partial\bar{M}}{\partial a}\bar{M}^{-1}$,

$$[I - \gamma\sigma(A - \alpha\nabla_A L(A, B))]^{-1} = [U + \alpha\gamma M + O(\alpha^2)]^{-1}$$

$$\begin{aligned}
&= \varphi(\alpha) \\
&= \varphi(0) + \frac{\partial \varphi(0)}{\partial \alpha}\alpha + O(\alpha^2) \\
&= U^{-1} - \alpha\gamma U^{-1} M U^{-1} + 2\alpha O(\alpha) + O(\alpha^2) \\
&= U^{-1} - \alpha\gamma U^{-1} M U^{-1} + O(\alpha^2)
\end{aligned}$$

Summarizing above,

$$\begin{aligned}
&[I_m - \gamma\sigma(A - \alpha\nabla_A L(A,B))]^{-1} \\
&= U^{-1} - \alpha\gamma U^{-1} [J_1 \nabla_A L(A,B)_{*1} \quad \cdots \quad J_m \nabla_A L(A,B)_{*m}] U^{-1} + O(\alpha^2)
\end{aligned} \tag{25}$$

### A.1.5 Putting Results Together for Induced Dynamics

We now consider the dynamics of $Z_t := B_t U_t^{-1}$ in $\mathbb{R}^{m_y \times m}$ that is induced by the gradient dynamics of $(A_t, B_t)$:

$$\frac{d}{dt}A_t = -\frac{\partial L}{\partial A}(A_t, B_t), \quad \frac{d}{dt}B_t = -\frac{\partial L}{\partial B}(A_t, B_t), \quad \forall t \ge 0.$$

Continuing the previous subsection, we first consider the dynamics discretized by the Euler method:

$$Z_{t+1} := B_{t+1} U_{t+1}^{-1} = [B_t - \alpha\nabla_B L(A_t, B_t)][I_m - \gamma\sigma(A_t - \alpha\nabla_A L(A_t, B_t))]^{-1},$$

where $\alpha > 0$. Then, substituting (25) into the right-hand side of this equation,

$$\begin{aligned}
&Z_{t+1} \\
&= B_{t+1}[U_t^{-1} - \alpha\gamma U_t^{-1}[J_{1,t}\nabla_A L(A_t, B_t)_{*1} \quad \cdots \quad J_{m,t}\nabla_A L(A_t, B_t)_{*m}] U_t^{-1} + O(\alpha^2)] \\
&= B_{t+1}U_t^{-1} - \alpha\gamma B_{t+1}U_t^{-1}[J_{1,t}\nabla_A L(A_t, B_t)_{*1} \quad \cdots \quad J_{m,t}\nabla_A L(A_t, B_t)_{*m}] U_t^{-1} + O(\alpha^2).
\end{aligned}$$

Using $B_{t+1} = [B_t - \alpha\nabla_B L(A_t, B_t)]$, we have

$$\begin{aligned}
&\alpha\gamma B_{t+1}U_t^{-1}[J_{1,t}\nabla_A L(A_t, B_t)_{*1} \quad \cdots \quad J_{m,t}\nabla_A L(A_t, B_t)_{*m}] U_t^{-1} \\
&= \alpha\gamma B_t U_t^{-1}[J_{1,t}\nabla_A L(A_t, B_t)_{*1} \quad \cdots \quad J_{m,t}\nabla_A L(A_t, B_t)_{*m}] U_t^{-1} + O(\alpha^2) \\
&= \alpha\gamma Z_t [J_{1,t}\nabla_A L(A_t, B_t)_{*1} \quad \cdots \quad J_{m,t}\nabla_A L(A_t, B_t)_{*m}] U_t^{-1} + O(\alpha^2).
\end{aligned}$$

Since $(U^{-\top})_{*k} = ((U^{-1})^\top)_{*k} = ((U^{-1})_{k*})^\top$, $U^{-1} = \begin{bmatrix} (U^{-1})_{1*} \\ \vdots \\ (U^{-1})_{m*} \end{bmatrix}$ and $\nabla_A L(A,B)_{*k} =$

$\left(\frac{\partial L(A,B)}{\partial A}\right)_{*k} = \gamma J_k^\top (BU^{-1})^\top \nabla L_0(BU^{-1})(U^{-\top})_{*k}$ from (24), we have that

$$\begin{aligned}
&Z_t [J_{1,t}\nabla_A L(A_t, B_t)_{*1} \quad \cdots \quad J_{m,t}\nabla_A L(A_t, B_t)_{*m}] U_t^{-1} \\
&= \gamma Z_t \left[ J_{1,t}J_{1,t}^\top Z_t^\top \nabla L_0(Z_t)(U_t^{-\top})_{*1} \quad \cdots \quad J_{m,t}J_{m,t}^\top Z_t^\top \nabla L_0(Z_t)(U_t^{-\top})_{*m} \right] U_t^{-1} \\
&= \gamma Z_t \left[ J_{1,t}J_{1,t}^\top Z_t^\top \nabla L_0(Z_t)(U_t^{-\top})_{*1} \quad \cdots \quad J_{m,t}J_{m,t}^\top Z_t^\top \nabla L_0(Z_t)(U_t^{-\top})_{*m} \right] \begin{bmatrix} (U_t^{-1})_{1*} \\ \vdots \\ (U_t^{-1})_{m*} \end{bmatrix} \\
&= \sum_{k=1}^m Z_t J_{k,t} J_{k,t}^\top Z_t^\top \nabla L_0(Z_t)((U_t^{-1})_{k*})^\top (U_t^{-1})_{k*} \\
&= \sum_{k=1}^m Z_t J_{k,t} J_{k,t}^\top Z_t^\top \nabla L(Z_t)(U_t^{-\top})_{*k}(U_t^{-1})_{k*}
\end{aligned}$$

On the other hand, using $B_{t+1} = [B_t - \alpha\nabla_B L(A_t, B_t)]$ and $\nabla_B L(A,B) := \frac{\partial L(A,B)}{\partial B} = \left(\sum_{i=1}^n \left(\frac{\partial \ell(\hat{y}_i, y_i)}{\partial \hat{y}_i}\right)^\top \phi(x_i)^\top\right) U^{-\top} = \nabla L_0(BU^{-1})U^{-\top}$, we have

$B_{t+1}U_t^{-1} = Z_t - \alpha \nabla L_0(Z_t)U_t^{-\top}U_t^{-1}$. Summarizing these equations by noticing $U^{-\top}U^{-1} = \sum_{k=1}^{m}(U^{-\top})_{*k}(U^{-1})_{k*}$ yields that

$$Z_{t+1} = Z_t - \alpha \nabla L_0(Z_t) \sum_{k=1}^{m}(U_t^{-\top})_{*k}(U_t^{-1})_{k*} - \alpha\gamma^2 \sum_{k=1}^{m} Z_t J_{k,t}J_{k,t}^{\top}Z_t^{\top}\nabla L_0(Z_t)(U_t^{-\top})_{*k}(U_t^{-1})_{k*} + O(\alpha^2)$$

$$= Z_t - \alpha \left( \sum_{k=1}^{m} \nabla L_0(Z_t)(U_t^{-\top})_{*k}(U_t^{-1})_{k*} + \gamma^2 Z_t J_{k,t}J_{k,t}^{\top}Z_t^{\top}\nabla L_0(Z_t)(U_t^{-\top})_{*k}(U_t^{-1})_{k*} \right) + O(\alpha^2)$$

$$= Z_t - \alpha \left( \sum_{k=1}^{m} (I_{m_y} + \gamma^2 Z_t J_{k,t}J_{k,t}^{\top}Z_t^{\top})\nabla L_0(Z_t)(U_t^{-\top})_{*k}(U_t^{-1})_{k*} \right) + O(\alpha^2)$$

By vectorizing both sides,

$\mathrm{vec}(Z_{t+1})$

$$= \mathrm{vec}(Z_t) - \alpha \left( \sum_{k=1}^{m} \mathrm{vec}\left((I_{m_y} + \gamma^2 Z_t J_{k,t}J_{k,t}^{\top}Z_t^{\top})\nabla L_0(Z_t)(U_t^{-\top})_{*k}(U_t^{-1})_{k*}\right) \right) + O(\alpha^2)$$

$$= \mathrm{vec}(Z_t) - \alpha \left( \sum_{k=1}^{m} [(U_t^{-\top})_{*k}(U_t^{-1})_{k*} \otimes (I_{m_y} + \gamma^2 Z_t J_{k,t}J_{k,t}^{\top}Z_t^{\top})] \right) \mathrm{vec}(\nabla L_0(Z_t)) + O(\alpha^2)$$

By defining

$$D_t := \sum_{k=1}^{m} [(U_t^{-\top})_{*k}(U_t^{-1})_{k*} \otimes (I_{m_y} + \gamma^2 Z_t J_{k,t}J_{k,t}^{\top}Z_t^{\top})],$$

we have

$$\mathrm{vec}(Z_{t+1}) = \mathrm{vec}(Z_t) - \alpha D_t \mathrm{vec}(\nabla L_0(Z_t)) + O(\alpha^2).$$

This implies that

$$\frac{\mathrm{vec}(Z_{t+1}) - \mathrm{vec}(Z_t)}{\alpha} = -D_t \mathrm{vec}(\nabla L_0(Z_t)) + O(\alpha),$$

where $\alpha > 0$. By recalling the definition of the Euler method and defining $Z(t) = Z_t$, we can rewrite this as

$$\frac{\mathrm{vec}(Z(t+\alpha)) - \mathrm{vec}(Z(t))}{\alpha} = -D_t \mathrm{vec}(\nabla L_0(Z_t)) + O(\alpha).$$

By taking the limit for $\alpha \to 0$ and going back to continuous-time dynamics, this implies that

$$\frac{d}{dt} \mathrm{vec}(Z_t) = -D_t \mathrm{vec}(\nabla L_0(Z_t)). \tag{26}$$

Here, we note that the complex interaction over $m$ due to the nonlinearity is factorized out into the matrix $D_t$. Furthermore, the interaction within the matrix $D_t$ has more structures when compared with that in the gradients themselves from (24). For example, unlike the gradients, the interaction over $m$ even within $D_t$ can be factorized out in the case of $m_y = 1$ as:

$$D = \sum_{k=1}^{m} \left[ (U^{-\top})_{*k}(U^{-1})_{k*} \otimes \left( I_{m_y} + \gamma^2 Z J_k J_k^{\top} Z^{\top} \right) \right]$$

$$= \sum_{k=1}^{m} \left( 1 + \gamma^2 Z J_k J_k^{\top} Z^{\top} \right) (U^{-\top})_{*k}(U^{-1})_{k*}$$

$$= U^{-\top} \mathrm{diag}\left( \begin{bmatrix} 1 + \gamma^2 Z J_1 J_1^{\top} Z^{\top} \\ \vdots \\ 1 + \gamma^2 Z J_m J_m^{\top} Z^{\top} \end{bmatrix} \right) U^{-1}$$

$$= U^{-\top} \left( I_m + \mathrm{diag}\left( \begin{bmatrix} \gamma^2 Z J_1 J_1^{\top} Z^{\top} \\ \vdots \\ \gamma^2 Z J_m J_m^{\top} Z^{\top} \end{bmatrix} \right) \right) U^{-1}.$$

Although we do not assume $m_y = 1$, this illustrates the additional structure well.

### A.1.6 ANALYSIS OF THE MATRIX $D_t$

From the definition of $D_t$, we have that

$$
\begin{aligned}
D_t &= \sum_{k=1}^{m} \left[ (U_t^{-\top})_{*k} (U_t^{-1})_{k*} \otimes \left( I_{m_y} + \gamma^2 Z_t J_{k,t} J_{k,t}^\top Z_t^\top \right) \right] \\
&= \sum_{k=1}^{m} \left[ (U^{-\top})_{*k} (U^{-1})_{k*} \otimes I_{m_y} \right] + \sum_{k=1}^{m} \left[ (U^{-\top})_{*k} (U^{-1})_{k*} \otimes \gamma^2 Z_t J_{k,t} J_{k,t}^\top Z_t^\top \right] \\
&= \left[ \sum_{k=1}^{m} (U^{-\top})_{*k} (U^{-1})_{k*} \otimes I_{m_y} \right] + \sum_{k=1}^{m} \left[ (U^{-\top})_{*k} (U^{-1})_{k*} \otimes \gamma^2 Z_t J_{k,t} J_{k,t}^\top Z_t^\top \right] \\
&= \left[ U^{-\top} U^{-1} \otimes I_{m_y} \right] + \sum_{k=1}^{m} \left[ (U^{-\top})_{*k} (U^{-1})_{k*} \otimes \gamma^2 Z_t J_{k,t} J_{k,t}^\top Z_t^\top \right] \quad (27)
\end{aligned}
$$

Since $U^{-\top} U^{-1}$ is positive definite, $I_{m_y}$ is positive definite, and a Kronecker product of two positive definite matrices is positive definite (since the eigenvalues of Kronecker product are the products of eigenvalues of the two matrices), we have

$$
\left[ U^{-\top} U^{-1} \otimes I_{m_y} \right] \succ 0. \quad (28)
$$

Since $(U^{-\top})_{*k} (U^{-1})_{k*}$ is positive semidefinite, $\gamma^2 Z_t J_{k,t} J_{k,t}^\top Z_t^\top$ is positive semidefinite, and a Kronecker product of two positive semidefinite matrices is positive semidefinite (since the eigenvalues of Kronecker product are the products of eigenvalues of the two matrices), we have

$$
\left[ (U^{-\top})_{*k} (U^{-1})_{k*} \otimes \gamma^2 Z_t J_{k,t} J_{k,t}^\top Z_t^\top \right] \succeq 0.
$$

Since a sum of positive semidefinite matrices is positive semidefinite (from the definition of positive semi-definiteness: $x^\top M_k x \geq 0 \Rightarrow x^\top (\sum_k M_k) x = \sum_k x^\top M_k x \geq 0$),

$$
\sum_{k=1}^{m} \left[ (U^{-\top})_{*k} (U^{-1})_{k*} \otimes \gamma^2 Z_t J_{k,t} J_{k,t}^\top Z_t^\top \right] \succeq 0. \quad (29)
$$

Since a sum of a positive *definite* matrix and positive *semidefinite* matrix is positive *definite* (from the definition of positive definiteness and positive definiteness: $(x^\top M_1 x > 0 \wedge x^\top M_2 x) \Rightarrow x^\top (M_1 + M_2) x = x^\top M_1 x + x^\top M_2 x > 0$),

$$
\begin{aligned}
D_t &= \sum_{k=1}^{m} \left[ (U^{-\top})_{*k} (U^{-1})_{k*} \otimes \left( I_{m_y} + \gamma^2 Z_t J_{k,t} J_{k,t}^\top Z_t^\top \right) \right] \\
&= \left[ U^{-\top} U^{-1} \otimes I_{m_y} \right] + \sum_{k=1}^{m} \left[ (U^{-\top})_{*k} (U^{-1})_{k*} \otimes \gamma^2 Z_t J_{k,t} J_{k,t}^\top Z_t^\top \right] \succ 0.
\end{aligned}
$$

Therefore, $D_t$ is a positive definite matrix for any $t$ and hence

$$
\lambda_T := \inf_{t \in [0,T]} \lambda_{\min}(D_t) > 0. \quad (30)
$$

### A.1.7 CONVERGENCE RATE VIA POLYAK-ŁOJASIEWICZ INEQUALITY AND NORM BOUNDS

Let $R \in (0, \infty]$ and $T > 0$ be arbitrary. By taking derivative of $L_0(Z_t) - L_{0,R}^*$ with respect to time $t$ with $Z_t := B_t U_t^{-1}$,

$$
\begin{aligned}
\frac{d}{dt} \left( L_0(Z_t) - L_{0,R}^* \right) &= \left( \sum_{i=1}^{m_y} \sum_{j=1}^{m} \left( \frac{dL_0}{dW}(Z_t) \right)_{ij} \left( \frac{d}{dt}(Z_t) \right)_{ij} \right) - \frac{d}{dt} L^*, \\
&= \sum_{i=1}^{m_y} \sum_{j=1}^{m} \left( \frac{dL_0}{dW}(Z_t) \right)_{ij} \left( \frac{d}{dt}(Z_t) \right)_{ij}
\end{aligned}
$$

where we used the chain rule and the fact that $\frac{d}{dt}L^*_{0,R} = 0$. By using the vectorization notation with $\nabla L_0(Z_t) = \frac{dL_0}{dW}(Z_t)$,

$$\frac{d}{dt}\left(L_0(Z_t) - L^*_{0,R}\right) = \text{vec}\left[\nabla L_0(Z_t)\right]^\top \text{vec}\left[\frac{d}{dt}(Z_t)\right],$$

By using (26) for the equation of $\text{vec}\left[\frac{d}{dt}(Z_t)\right]$,

$$\frac{d}{dt}\left(L_0(Z_t) - L^*_{0,R}\right) = -\text{vec}\left[\nabla L_0(Z_t)\right]^\top D_t \text{vec}[\nabla L_0(Z_t)]$$

$$\leq -\lambda_{\min}(D_t)\|\text{vec}\left[\nabla L_0(Z_t)\right]\|^2_2$$

$$= -\lambda_{\min}(D_t)\|\nabla L_0(Z_t)\|^2_F$$

Using the condition that $\nabla L_0$ satisfies the the Polyak-Łojasiewicz inequality with radius $R$, if $\|Z_t\|_1 < R$, then we have that for all $t \in [0, T]$,

$$\frac{d}{dt}\left(L_0(Z_t) - L^*_{0,R}\right) \leq -2\kappa\lambda_{\min}(D_t)(L_0(Z_t) - L^*_{0,R})$$

$$\leq -2\kappa\lambda_T(L_0(Z_t) - L^*_{0,R}).$$

By solving the differential equation, this implies that if $\|Z_t\|_1 < R$,

$$L_0(Z_T) - L^*_{0,R} \leq \left(L_0(Z_0) - L^*_{0,R}\right)e^{-2\kappa\lambda_T T},$$

Since $L(A_t, B_t) = L_0(Z_t)$, if $\|Z_t\|_1 < R$,

$$L(A_T, B_T) \leq L^*_{0,R} + (L(A_0, B_0) - L^*_{0,R})e^{-2\kappa\lambda_T T}. \tag{31}$$

We now complete the proof of the first part of the desired statement by showing that $\|B\|_1 < (1 - \gamma)R$ implies $\|Z_t\|_1 < R$. With $Z = BU^{-1}$, since any induced operator norm is a submultiplicative matrix norm,

$$\|Z\|_1 = \|B(I_m - \gamma\sigma(A))^{-1}\|_1 \leq \|B\|_1\|(I_m - \gamma\sigma(A))^{-1}\|_1.$$

We can then rewrite

$$\|(I_m - \gamma\sigma(A))^{-1}\|_1 = \|((I_m - \gamma\sigma(A))^{-1})^\top\|_\infty = \|(I_m - \gamma\sigma(A)^\top)^{-1}\|_\infty.$$

Here, the matrix $I_m - \gamma\sigma(A)^\top$ is strictly diagonally dominant: i.e., $|I_m - \gamma\sigma(A)^\top|_{ii} > \sum_{j\neq i}|I_m - \gamma\sigma(A)|_{ij}$ for any $i$. This can be shown as follows: for any $j$,

$$1 > \gamma \iff 1 > \gamma\sum_i \sigma(A)_{ij}$$

$$\iff 1 > \gamma\sigma(A)_{jj} + \sum_{i\neq j}\gamma\sigma(A)_{ij}$$

$$\iff 1 - \gamma\sigma(A)_{jj} > \sum_{i\neq j}\gamma\sigma(A)_{ij}$$

$$\iff |I_m - \gamma\sigma(A)|_{jj} > \sum_{i\neq j}|-\gamma\sigma(A)|_{ij}$$

$$\iff |I_m - \gamma\sigma(A)|_{jj} > \sum_{i\neq j}|I_m - \gamma\sigma(A)|_{ij}$$

$$\iff |I_m - \gamma\sigma(A)^\top|_{jj} > \sum_{i\neq j}|I_m - \gamma\sigma(A)^\top|_{ji}$$

This calculation also shows that $|I_m - \gamma\sigma(A)|_{jj} - \sum_{i\neq j}|I_m - \gamma\sigma(A)|_{ij} = 1 - \gamma$ for all $j$. Thus, using the Ahlberg–Nilson–Varah bound for the strictly diagonally dominant matrix (Ahlberg & Nilson, 1963; Varah, 1975; Morača, 2008), we have

$$\|(I_m - \gamma\sigma(A)^\top)^{-1}\|_\infty \leq \frac{1}{\min_j(|I_m - \gamma\sigma(A)|_{jj} - \sum_{i\neq j}|I_m - \gamma\sigma(A)|_{ij})} = \frac{1}{1 - \gamma}.$$

By taking transpose,

$$\|(I_m - \gamma\sigma(A))^{-1}\|_1 \leq \frac{1}{1-\gamma}.$$

Summarizing above,

$$\|Z\|_1 = \|B(I_m - \gamma\sigma(A))^{-1}\|_1 \leq \|B\|_1 \frac{1}{1-\gamma}.$$

Therefore, if $\|B\|_1 < R(1-\gamma)$, then $\|Z\|_1 = \|B(I_m - \gamma\sigma(A))^{-1}\|_1 \leq \|B\|_1 \frac{1}{1-\gamma} < R$, as desired. Combining this with (31) implies that if $\|B\|_1 < R(1-\gamma)$,

$$L(A_t, B_t) \leq L_{0,R}^* + (L(A_0, B_0) - L_{0,R}^*)e^{-2\kappa\lambda_T T}.$$

Recall that $L_{0,R}^* = \inf_{W:\|W\|_1 < R} L_0(W)$ and $L_R^* = \inf_{A \in \mathbb{R}^{m \times m}, B \in \mathcal{B}_R} L(A, B)$ where $\mathcal{B}_R = \{B \in \mathbb{R}^{m_y \times m} \mid \|B\|_1 < (1-\gamma)R\}$. Here, $B \in \mathcal{B}_R$ implies that $\|Z\|_1 = \|BU^{-1}\|_1 \leq \|B\|_1\|U^{-1}\|_1 < (1-\gamma)R\|U^{-1}\|_1 \leq R$, using the above upper bond $\|U^{-1}\|_1 = \|(I_m - \gamma\sigma(A))^{-1}\|_1 \leq \frac{1}{1-\gamma}$. Since $L(A, B) = L_0(Z)$ with $Z = BU^{-1}$, this implies that $L_{0,R}^* \leq L_R^*$ and thus

$$L(A_t, B_t) \leq L_{0,R}^* + (L(A_0, B_0) - L_{0,R}^*)e^{-2\kappa\lambda_T T} \leq L_R^* + (L(A_0, B_0) - L_{0,R}^*)e^{-2\kappa\lambda_T T}.$$

This completes the first part of the desired statement of Theorem 1.

The remaining task is to lower bound $\lambda_T$, which is completed as follows: for any $(A, B)$,

$$
\begin{aligned}
\lambda_{\min}(D) &= \min_{v:\|v\|=1} v^\top D v \\
&= \min_{v:\|v\|=1} v^\top \left( \sum_{k=1}^m [(U_t^{-\top})_{*k}(U_t^{-1})_{k*} \otimes (I_{m_y} + \gamma^2 Z_t J_{k,t} J_{k,t}^\top Z_t^\top)] \right) v \\
&\geq \min_{v:\|v\|=1} v^\top \left[ U^{-\top}U^{-1} \otimes I_{m_y} \right] v \\
&= \lambda_{\min}\left( \left[ U^{-\top}U^{-1} \otimes I_{m_y} \right] \right) \\
&= \lambda_{\min}(U^{-\top}U^{-1}) \\
&= \sigma_{\min}^2(U^{-1}) = \frac{1}{\|U\|_2^2} \geq \frac{1}{m\|U\|_1^2}
\end{aligned}
\tag{32}
$$

where the third line follows from (27)–(29), the fifth line follows from the property of Kronecker product (the eigenvalues of Kronecker product of two matrices are the products of eigenvalues of the two matrices), and the last inequality follows from the relation between the spectral norm and the norm $\|\cdot\|_1$. We now compute $\|U\|_1$ as: for any $(A, B)$,

$$
\begin{aligned}
\|U\|_1 &= \|I_m - \gamma\sigma(A)\|_1 \\
&= \max_j \sum_i |(I_m - \gamma\sigma(A))_{ij}| \\
&= \max_j \sum_i |(I_m)_{ij} - \gamma\sigma(A)_{ij}| \\
&= \max_j |(I_m)_{jj} - \gamma\sigma(A)_{jj}| + \sum_{i \neq j} |(I_m)_{ij} - \gamma\sigma(A)_{ij}| \\
&= \max_j |1 - \gamma\sigma(A)_{jj}| + \sum_{i \neq j} |-\gamma\sigma(A)_{ij}| \\
&= \max_j 1 - \gamma\sigma(A)_{jj} + \sum_{i \neq j} \gamma\sigma(A)_{ij} \\
&= \max_j 1 + \gamma\left( \left( \sum_{i \neq j} \sigma(A)_{ij} \right) - \sigma(A)_{jj} \right) \\
&\leq 1 + \gamma.
\end{aligned}
$$

By substituting this into (32), we have that for any $(A, B)$ (and hence for any $t$),

$$\lambda_{\min}(D) \geq \frac{1}{m(1+\gamma)^2}. \tag{33}$$

This completes the proof for both the first and second parts of the statement of Theorem 1.

$\square$

### A.2 PROOF OF THEOREM 2

We first show that with $\delta_t > 0$ sufficiently small, we have $G_t \succ 0$. Recall that

$$G_t = U_t \left( S_t^{-1} - \delta_t F_t \right) U_t^\top = U_t S_t^{-1} U_t^\top - \delta_t U_t F_t U_t^\top.$$

Thus, with $\delta_t > 0$ sufficiently small, for any $v \neq 0$,

$$v^\top G v = v^\top U_t S_t^{-1} U_t^\top v - \delta_t v^\top U_t F_t U_t^\top v,$$

which is dominated by the first term $v^\top U_t S_t^{-1} U_t^\top v$ if the matrix $U_t S_t^{-1} U_t^\top$ is positive definite. Since $S_t := I_m + \gamma^2 \operatorname{diag}(v_t^S)$ with $v_t^S \in \mathbb{R}^m$ and $(v_t^S)_k := \|J_{k,t}^\top (B_t U_t^{-1})^\top\|_2^2$ for $k = 1, 2, \ldots, m$, the matrix $U_t S_t^{-1} U_t^\top$ is positive definite. Thus, with $\delta_t > 0$ sufficiently small, $v^\top G v$ is dominated by the first term, which is positive (since $U_t S_t^{-1} U_t^\top$ is positive definite), and thus we have $G_t \succ 0$.

Then we observe that the output of $\operatorname{argmin}_{v:\|v\|_{G_t} \leq \delta_t \|\frac{d}{dt} \operatorname{vec}(B_t U_t^{-1})\|_{G_t}} L_0^t(v)$ is the set of solutions of the following constrained optimization problem:

$$\underset{v}{\operatorname{minimize}} \quad L_0^t(v) \quad \text{s.t.} \quad \|v\|_{G_t}^2 - \delta_t^2 \left\| \frac{d}{dt} \operatorname{vec}(B_t U_t^{-1}) \right\|_{G_t}^2 \leq 0.$$

Since this optimization problem is convex, one of the sufficient conditions for global optimality is the KKT condition with a multiplier $\mu \in \mathbb{R}$:

$$\nabla L_0^t(v) + 2\mu G_t v = 0$$
$$\mu \geq 0$$
$$\mu \left( \|v\|_{G_t}^2 - \delta_t^2 \left\| \frac{d}{dt} \operatorname{vec}(B_t U_t^{-1}) \right\|_{G_t}^2 \right) = 0.$$

Therefore, the desired statement is obtained if the above KKT condition is satisfied by $v = \delta_t \left( \frac{d}{dt} \operatorname{vec}(B_t U_t^{-1}) \right)$ with some multiplier $\mu$. The rest of this proof shows that the KKT condition is satisfied by setting $v = \delta_t \left( \frac{d}{dt} \operatorname{vec}(B_t U_t^{-1}) \right)$ and $\mu = \frac{1}{2\delta_t}$. With this choice, the last two conditions of the KKT condition hold, since

$$\mu = \frac{1}{2\delta_t} \geq 0,$$

and

$$\|v\|_{G_t}^2 - \delta_t^2 \left\| \frac{d}{dt} \operatorname{vec}(B_t U_t^{-1}) \right\|_{G_t}^2 = \delta_t^2 \left\| \frac{d}{dt} \operatorname{vec}(B_t U_t^{-1}) \right\|_{G_t}^2 - \delta_t^2 \left\| \frac{d}{dt} \operatorname{vec}(B_t U_t^{-1}) \right\|_{G_t}^2 = 0.$$

The remaining task is to show that $\nabla L_0^t(v) + 2\mu G_t v = 0$ with $v = \delta_t \left( \frac{d}{dt} \operatorname{vec}(B_t U_t^{-1}) \right)$ and $\mu = \frac{1}{2\delta_t}$. From the definition of $L_0^t$,

$$\nabla L_0^t(v) + 2\mu G_t v = \nabla L_0^{\operatorname{vec}}(\operatorname{vec}(B_t U_t^{-1})) + \nabla^2 L_0^{\operatorname{vec}}(\operatorname{vec}(B_t U_t^{-1}))v + 2\mu G_t v. \tag{34}$$

We now compute and $\nabla L_0^{\operatorname{vec}}$ and $\nabla^2 L_0^{\operatorname{vec}}$. Since $\nabla L_0(W) := \frac{\partial L_0(W)}{\partial W} = \sum_{i=1}^n \left( \frac{\partial \ell(\hat{y}_i, y_i)}{\partial \hat{y}_i} \right)^\top x_i^\top$,

$$\operatorname{vec}(\nabla L_0(W)) = \sum_{i=1}^n \operatorname{vec} \left( I_{m_y} \left( \frac{\partial \ell(\hat{y}_i, y_i)}{\partial \hat{y}_i} \right)^\top \phi(x_i)^\top \right) = \sum_{i=1}^n [\phi(x_i) \otimes I_{m_y}] \left( \frac{\partial \ell(\hat{y}_i, y_i)}{\partial \hat{y}_i} \right)^\top,$$

where

$$\hat{y}_i := W\phi(x_i) = [\phi(x_i)^\top \otimes I_{m_y}]\,\mathrm{vec}[W].$$

Therefore,

$$\nabla L_0^{\mathrm{vec}}(\mathrm{vec}(W)) = \mathrm{vec}(\nabla L_0(W)) = \sum_{i=1}^{n}[\phi(x_i) \otimes I_{m_y}]\left(\frac{\partial \ell(\hat{y}_i, y_i)}{\partial \hat{y}_i}\right)^\top.$$

For the Hessian,

$$\begin{aligned}
\nabla^2 L_0^{\mathrm{vec}}(\mathrm{vec}(W)) &= \frac{\partial}{\partial\,\mathrm{vec}(W)}\nabla L_0^{\mathrm{vec}}(\mathrm{vec}(W)) \\
&= \sum_{i=1}^{n}[\phi(x_i) \otimes I_{m_y}]\left(\frac{\partial}{\partial \hat{y}_i}\left(\frac{\partial \ell(\hat{y}_i, y_i)}{\partial \hat{y}_i}\right)^\top\right)\frac{\partial \hat{y}_i}{\partial\,\mathrm{vec}(W)} \\
&= \sum_{i=1}^{n}[\phi(x_i) \otimes I_{m_y}]\left(\frac{\partial}{\partial \hat{y}_i}\left(\frac{\partial \ell(\hat{y}_i, y_i)}{\partial \hat{y}_i}\right)^\top\right)[\phi(x_i)^\top \otimes I_{m_y}]
\end{aligned}$$

By defining $\ell_i(z) = \ell(z, y_i)$ and $\nabla^2 \ell_i(z) = \frac{\partial}{\partial z}\left(\frac{\partial \ell_i(z)}{\partial z}\right)^\top$,

$$\nabla^2 L_0^{\mathrm{vec}}(\mathrm{vec}(W)) = \sum_{i=1}^{n}[\phi(x_i) \otimes I_{m_y}]\nabla^2 \ell_i(W\phi(x_i))[\phi(x_i)^\top \otimes I_{m_y}]. \tag{35}$$

Since we have that

$$\begin{aligned}
&(I_m - \gamma\sigma(A))\left(\sum_{k=0}^{l}\gamma^k\sigma(A)^k\right) \\
&= I_m - \gamma\sigma(A) + \gamma\sigma(A) - (\gamma\sigma(A))^2 + (\gamma\sigma(A))^2 - (\gamma\sigma(A))^3 + \cdots - (\gamma\sigma(A))^{l+1} \\
&= I - (\gamma\sigma(A))^{l+1},
\end{aligned}$$

we can write:

$$\begin{aligned}
(I_m - \gamma\sigma(A))\left(\lim_{l\to\infty} z^{(l)}(x, A)\right) &= \lim_{l\to\infty}(I_m - \gamma\sigma(A))\left(\sum_{k=0}^{l}\gamma^k\sigma(A)^k\phi(x)\right) \\
&= \left(I_m - \lim_{l\to\infty}(\gamma\sigma(A))^{l+1}\right)\phi(x) \\
&= \phi(x),
\end{aligned}$$

where we used the fact that $\gamma\sigma(A)_{ij} \geq 0$, $\|\sigma(A)\|_1 = \max_j \sum_i |\sigma(A)_{ij}| = 1$, and thus $\|\gamma\sigma(A)\|_1 < 1$ for any $\gamma \in (0, 1)$. This shows that $\left(\lim_{l\to\infty} z^{(l)}(x, A)\right) = z^*(x, A) = U^{-1}\phi(x)$, from which we have $\phi(x_i) = Uz^*(x_i, A)$.

Substituting $\phi(x_i) = Uz^*(x_i, A)$ into (35),

$$\begin{aligned}
&\nabla^2 L_0^{\mathrm{vec}}(\mathrm{vec}(W)) \\
&= \sum_{i=1}^{n}[Uz^*(x_i, A) \otimes I_{m_y}]\nabla^2 \ell_i(W\phi(x_i))[z^*(x_i, A)^\top U^\top \otimes I_{m_y}]. \\
&= \sum_{i=1}^{n}[U \otimes I_{m_y}][z^*(x_i, A) \otimes I_{m_y}]\nabla^2 \ell_i(W\phi(x_i))[z^*(x_i, A)^\top \otimes I_{m_y}][U^\top \otimes I_{m_y}]. \\
&= [U \otimes I_{m_y}]\left(\sum_{i=1}^{n}[z^*(x_i, A) \otimes I_{m_y}]\nabla^2 \ell_i(W\phi(x_i))[z^*(x_i, A)^\top \otimes I_{m_y}]\right)[U^\top \otimes I_{m_y}].
\end{aligned}$$

In the case of $m_y = 1$, since $I_{m_y} = 1$, $\nabla^2 L_0^{\text{vec}}(\text{vec}(W))$ is further simplified to:

$$\nabla^2 L_0^{\text{vec}}(\text{vec}(W)) = U \left( \sum_{i=1}^n \nabla^2 \ell_i(W\phi(x_i)) z^*(x_i, A) z^*(x_i, A)^\top \right) U^\top.$$

Therefore,

$$\nabla^2 L_0^{\text{vec}}(\text{vec}(B_t U_t^{-1})) = U_t F_t U_t^\top,$$

where

$$F_t = \sum_{i=1}^n \nabla^2 \ell_i(B_t U_t^{-1}\phi(x_i)) z^*(x_i, A_t) z^*(x_i, A_t)^\top.$$

By plugging $\mu = \frac{1}{2\delta_t}$ and $\nabla^2 L_0^{\text{vec}}(\text{vec}(B_t U_t^{-1})) = U_t F_t U_t^\top$ into (34),

$$\nabla L_0^t(v) + 2\mu G_t v = \nabla L_0^{\text{vec}}(\text{vec}(B_t U_t^{-1})) + U_t F_t U_t^\top v + \frac{1}{\delta_t} U_t \left( S_t^{-1} - \delta_t F_t \right) U_t^\top v$$

$$= \nabla L_0^{\text{vec}}(\text{vec}(B_t U_t^{-1})) + \frac{1}{\delta_t} U_t S_t^{-1} U_t^\top v.$$

By using $v = \delta_t \left( \frac{d}{dt} \text{vec}(B_t U_t^{-1}) \right)$,

$$\nabla L_0^t(v) + 2\mu G_t v = \nabla L_0^{\text{vec}}(\text{vec}(B_t U_t^{-1})) + U_t S_t^{-1} U_t^\top \left( \frac{d}{dt} \text{vec}(B_t U_t^{-1}) \right).$$

By plugging (26) into $\frac{d}{dt} \text{vec}(B_t U_t^{-1})$ with $Z_t = B_t U_t^{-1}$,

$$\nabla L_0^t(v) + 2\mu G_t v = \nabla L_0^{\text{vec}}(\text{vec}(B_t U_t^{-1})) - U_t S_t^{-1} U_t^\top D_t \text{vec}(\nabla L_0(Z_t)).$$

Recall that

$$D_t = \sum_{k=1}^m [(U_t^{-\top})_{*k}(U_t^{-1})_{k*} \otimes (I_{m_y} + \gamma^2 Z_t J_{k,t} J_{k,t}^\top Z_t^\top)].$$

In the case of $m_y = 1$, the matrix $D_t$ can be simplified as:

$$D = \sum_{k=1}^m \left[ (U^{-\top})_{*k}(U^{-1})_{k*} \otimes \left( I_{m_y} + \gamma^2 Z J_k J_k^\top Z^\top \right) \right]$$

$$= \sum_{k=1}^m \left( 1 + \gamma^2 Z J_k J_k^\top Z^\top \right) (U^{-\top})_{*k}(U^{-1})_{k*}$$

$$= U^{-\top} \text{diag} \left( \begin{bmatrix} 1 + \gamma^2 Z J_1 J_1^\top Z^\top \\ \vdots \\ 1 + \gamma^2 Z J_m J_m^\top Z^\top \end{bmatrix} \right) U^{-1}$$

$$= U^{-\top} S U^{-1}.$$

Plugging this into the above equation for $D_t$,

$$\nabla L_0^t(v) + 2\mu G_t v = \nabla L_0^{\text{vec}}(\text{vec}(B_t U_t^{-1})) - U_t S_t^{-1} U_t^\top U_t^{-\top} S_t U_t^{-1} \text{vec}(\nabla L_0(Z_t))$$

$$= \nabla L_0^{\text{vec}}(\text{vec}(B_t U_t^{-1})) - \nabla L_0^{\text{vec}}(\text{vec}(B_t U_t^{-1})) = 0.$$

Therefore, the constrained optimization problem at time $t$ is solved by

$$v = \delta_t \left( \frac{d}{dt} \text{vec}(B_t U_t^{-1}) \right),$$

which implies that

$$\frac{d}{dt} \text{vec}(B_t U_t^{-1}) = \frac{1}{\delta_t} \text{vec}(V_t), \quad \text{vec}(V_t) \in \underset{v: \|v\|_{G_t} \leq \delta_t \|\frac{d}{dt} \text{vec}(B_t U_t^{-1})\|_{G_t}}{\text{argmin}} L_0^t(v),$$

By multiplying $\phi(x)^\top \otimes I_{m_y}$ to each side of the equation, we have

$$\frac{d}{dt}[\phi(x)^\top \otimes I_{m_y}]\operatorname{vec}(B_t U_t^{-1}) = B_t\left(\lim_{l\to\infty} z^{(l)}(x, A_t)\right),$$

$$\frac{1}{\delta_t}[\phi(x)^\top \otimes I_{m_y}]\operatorname{vec}(V_t) = \frac{1}{\delta_t}V_t\phi(x),$$

yielding that

$$\frac{d}{dt}B_t\left(\lim_{l\to\infty} z^{(l)}(x, A_t)\right) = \frac{1}{\delta_t}V_t\phi(x).$$

This proves the desired statement of Theorem 2.

$\square$

### A.3 PROOF OF COROLLARY 1

The assumption $\operatorname{rank}(\Phi) = \min(n, m)$ implies that $\sigma_{\min}(\Phi) > 0$. Moreover, the square loss $\ell$ satisfies the assumption of the differentiability. Thus, Theorem 1 implies the statement of this corollary if $L_0$ with the square loss satisfies the Polyak-Łojasiewicz inequality for any $W \in \mathbb{R}^{m_y \times m}$ with parameter $\kappa = 2\sigma_{\min}(\Phi)^2$. This is to be shown in the rest of this proof. By setting $\varphi = L_0$ in Definition 1, we have $\|\nabla \varphi_{\operatorname{vec}}(\operatorname{vec}(q))\|_2^2 = \|\nabla L_0(W)\|_F^2$. With the square loss, we can write $L_0(W) = \sum_{i=1}^n \|W\phi(x_i) - y_i\|_2^2 = \|W\Phi - Y\|_F^2$ where $\Phi \in \mathbb{R}^{m \times n}$ and $Y \in \mathbb{R}^{m_y \times n}$ with $\Phi_{ji} = \phi(x_i)_j$ and $Y_{ji} = (y_i)_j$. Thus,

$$\nabla L_0(W) = 2(W\Phi - Y)\Phi^\top \in \mathbb{R}^{m_y \times m}.$$

We first consider the case of $m \le n$. In this case, we consider the vectorization $L_0^{\operatorname{vec}}(\operatorname{vec}(W)) = L_0(W)$ and derive the gradient with respect to $\operatorname{vec}(W)$:

$$\nabla L_0^{\operatorname{vec}}(\operatorname{vec}(W)) = 2\operatorname{vec}((W\Phi - Y)\Phi^\top) = 2[\Phi \otimes I_{m_y}]\operatorname{vec}(W\Phi - Y).$$

Then, the Hessian can be easily computed as

$$\nabla^2 L_0^{\operatorname{vec}}(\operatorname{vec}(W)) = 2[\Phi \otimes I_{m_y}][\Phi^\top \otimes I_{m_y}] = 2[\Phi\Phi^\top \otimes I_{m_y}],$$

where $I_{m_y}$ is the identity matrix of size $m_y$ by $m_y$. Since the singular values of Kronecker product of the two matrices is the product of singular values of each matrix, we have

$$\nabla^2 L_0^{\operatorname{vec}}(\operatorname{vec}(W)) \succeq 2\sigma_{\min}(\Phi)^2 I_{m_y m},$$

where we used the fact that $m \le n$ in this case. Since $W$ is arbitrary, this implies that $L_0^{\operatorname{vec}}$ is strongly convex with parameter $2\sigma_{\min}(\Phi)^2 > 0$ in $\mathbb{R}^{m_y \times m}$. Since a strongly convex function with some parameter satisfies the Polyak-Łojasiewicz inequality with the same parameter (Karimi et al., 2016), this implies that $L_0^{\operatorname{vec}}$ (and hence $L_0$) satisfies the Polyak-Łojasiewicz inequality with parameter $2\sigma_{\min}(\Phi)^2 > 0$ in $\mathbb{R}^{m_y \times m}$ in the case of $m \le n$.

We now consider the remaining case of $m > n$. In this case, using the singular value decomposition of $\Phi = U\Sigma V^\top$,

$$\begin{aligned}
\frac{1}{2}\|\nabla L_0(W)\|_F^2 &= 2\|\Phi(W\Phi - Y)^\top\|_F^2 \\
&= 2\|U\Sigma V^\top(W\Phi - Y)^\top\|_F^2 \\
&= 2\|\Sigma V^\top(W\Phi - Y)^\top\|_F^2 \\
&\ge 2\sigma_{\min}(\Phi)^2\|V^\top(W\Phi - Y)^\top\|_F^2 \\
&= 2\sigma_{\min}(\Phi)^2 L_0(W) \\
&\ge 2\sigma_{\min}(\Phi)^2\left(L_0(W) - L_0^{**}\right)
\end{aligned}$$

for any $L_0^{**} \ge 0$, where the first line uses $\|q\|_F^2 = \|q^\top\|_F^2$, the second line uses the singular value decomposition, and the third and fourth line uses the fact that $U$ and $V$ are orthonormal matrices. The forth line uses the fact that $m > n$ in this case. Therefore, since $W$ is arbitrary, we have shown that $L_0$ satisfies the Polyak-Łojasiewicz inequality for any $W \in \mathbb{R}^{m_y \times m}$ with parameter $\kappa = 2\sigma_{\min}(\Phi)^2$ in both cases of $m > n$ and $m \le n$.

$\square$

### A.4 PROOF OF COROLLARY 2

The assumption $\mathrm{rank}(\Phi) = m$ implies that $\sigma_{\min}(\Phi) > 0$. Moreover, the logistic loss $\ell$ satisfies the assumption of the differentiability. Thus, Theorem 1 implies the statement of this corollary if $L_0$ with the logistic loss satisfies the Polyak-Łojasiewicz inequality with the given radius $R \in (0, \infty]$ and the parameter $\kappa = (2\tau + \rho(R))\sigma_{\min}(\Phi)^2$ where $\rho(R)$ depends on $R$. Let $R \in (0, \infty]$ be given. Note that we have $\rho(R) > 0$ if $R < \infty$, and $\rho(R) = 0$ if $R = \infty$. If $(R, \tau) = (\infty, 0)$, then $2\tau + \rho(R) = 0$ for which the statement of this corollary trivially holds (since the bound does not decrease). Therefore, we focus on the remaining case of $(R, \tau) \neq (\infty, 0)$. Since $(R, \tau) \neq (\infty, 0)$, we have $2\tau + \rho(R) > 0$. We first compute the Hessian with respect to $W$ as:

$$\nabla^2 L_0(W) = \sum_{i=1}^{n} \hat{p}_i(1 - \hat{p}_i)\phi(x_i)\phi(x_i)^\top + 2\tau \sum_{i=1}^{n} \phi(x_i)\phi(x_i)^\top,$$

where $\hat{p}_i = \frac{1}{1 + e^{-W\phi(x_i)}}$. Therefore,

$$
\begin{aligned}
v^\top \nabla^2 L_0(W)v &= \sum_{i=1}^{n} \hat{p}_i(1 - \hat{p}_i)v^\top \phi(x_i)\phi(x_i)^\top v + 2\tau v^\top \left( \sum_{i=1}^{n} \phi(x_i)\phi(x_i)^\top \right) v \\
&\geq \rho(R) \sum_{i=1}^{n} v^\top \phi(x_i)\phi(x_i)^\top v + 2\tau v^\top \left( \sum_{i=1}^{n} \phi(x_i)\phi(x_i)^\top \right) v \\
&= (2\tau + \rho(R))v^\top \Phi\Phi^\top v \\
&\geq (2\tau + \rho(R))\sigma_{\min}(\Phi)^2 > 0.
\end{aligned}
$$

Therefore, $L_0$ is strongly convex with parameter $(2\tau + \rho(R))\sigma_{\min}(\Phi)^2 > 0$. Since a strongly convex function with a parameter satisfies the Polyak-Łojasiewicz inequality with the same parameter (Karimi et al., 2016), this implies that $L_0$ satisfies the Polyak-Łojasiewicz inequality with the given radius $R \in (0, \infty]$ and parameter $(2\tau + \rho(R))\sigma_{\min}(\Phi)^2 > 0$. Since $R \in (0, \infty]$ is arbitrary, this implies the statement of this corollary by Theorem 1.

$\square$

### A.5 PROOF OF PROPOSITION 1

Let $(x, A)$ be given. By repeatedly applying the definition of $z^{(l)}(x, A)$, we obtain

$$
\begin{aligned}
z^{(l)}(x, A) &= \gamma\sigma(A)z^{(l-1)} + \phi(x) \\
&= \gamma\sigma(A)(\gamma\sigma(A)z^{(l-2)} + \phi(x)) + \phi(x) \\
&= \sum_{k=0}^{l} \gamma^k \sigma(A)^k \phi(x),
\end{aligned}
\tag{36}
$$

where $\sigma(A)^k$ represents the matrix multiplications of $k$ copies of the matrix $\sigma(A)$ with $\sigma(A)^0 = I_m$. In general, if $\sigma$ is identity, this sequence does not converge. However, with our definition of $\sigma$, we have

$$\|\sigma(A)\|_1 = \max_j \sum_i |\sigma(A)_{ij}| = 1.$$

Therefore, $\|\gamma\sigma(A)\|_1 = \gamma\|\sigma(A)\|_1 = \gamma < 1$ for any $\gamma \in (0, 1)$. Since an induced matrix norm is sub-multiplicative, this implies that

$$\|\gamma^k \sigma(A)^k\|_1 \leq \prod_{i=1}^{k} \|\gamma\sigma(A)\|_1 = \gamma^k.$$

In other words, each term $\|\gamma^k \sigma(A)^k\|_1$ in the series $\sum_{k=0}^{\infty} \|\gamma^k \sigma(A)^k\|_1$ is bounded by $\gamma^k$. Since the series $\sum_{k=0}^{\infty} \gamma^k$ converges in $\mathbb{R}$ with $\gamma \in (0, 1)$, this implies, by the comparison test, that the series $\sum_{k=0}^{\infty} \|\gamma^k \sigma(A)^k\|_1$ converges in $\mathbb{R}$. Thus, the sequence $(\sum_{k=0}^{l} \|\gamma^k \sigma(A)^k\|_1)_l$ is a Cauchy

sequence. By defining $s_l = \sum_{k=0}^{l} \gamma^k \sigma(A)^k$, we have $\|s_l - s_{l'}\|_1 = \|\sum_{k=l'+1}^{l} \gamma^k \sigma(A)^k\|_1 \leq \sum_{k=l'+1}^{l} \|\gamma^k \sigma(A)^k\|_1$ for any $l' > l$ by the triangle inequality of the (matrix) norm. Since $(\sum_{k=0}^{l} \|\gamma^k \sigma(A)^k\|_1)_l$ is a Cauchy sequence, this inequality implies that $(\sum_{k=0}^{l} \gamma^k \sigma(A)^k)_l$ is a Cauchy sequence (in a Banach space $(\mathbb{R}^{m \times m}, \|\cdot\|_1)$, which is isometric to $\mathbb{R}^{mm}$ under $\|\cdot\|_1$). Thus, the sequence $(\sum_{k=0}^{l} \gamma^k \sigma(A)^k)_l$ converges. From (36), this implies that the sequence $(z^{(l)}(x, A))_l$ converges. □

# B    ON THE IMPLICIT BIAS

In this section, we show that Theorem 2 suggests an implicit bias towards a simple function as a result of infinite depth, whereas understanding this bias in more details is left as an open problem. This section focuses on the case of the square loss $\ell(q, y_i) = \|q - y_i\|_2^2$ with $m_y = 1$. By solving $\text{vec}(V_t) \in \text{argmin}_{v \in \mathcal{V}} L_0^t(v)$ in Theorem 2 for the direction of the Newton method, Theorem 2 implies that

$$V_t = -r_t^\top (I_m - \gamma \sigma(A_t))^{-1} \in \mathbb{R}^{1 \times m}, \tag{37}$$

where $r_t \in \mathbb{R}^m$ is an error vector with each entry being a function of the residuals $f_{\theta_t}(x_i) - y_i$.

Since $\sigma(A_t)$ is a positive matrix and is a left stochastic matrix due to the nonlinearity $\sigma$, the Perron–Frobenius theorem (Perron, 1907; Frobenius, 1912) ensures that the largest eigenvalue of $\sigma(A_t)$ is one, any other eigenvalue in absolute value is strictly smaller than one, and any left eigenvector corresponding the largest eigenvalue is the vector $\eta \mathbf{1} = \eta[1, 1, \ldots, 1]^\top \in \mathbb{R}^m$ where $\zeta \in \mathbb{R}$ is some scalar. Thus, the largest eigenvalue of the matrix $(I_m - \gamma \sigma(A_t))^{-1}$ is $\frac{1}{1-\gamma}$, any other eigenvalue is in the form of $\frac{1}{1-\lambda_k \gamma}$ with $|\lambda_k| < 1$, and any left eigenvector corresponding the largest eigenvalue is $\eta \mathbf{1} \in \mathbb{R}^m$. By decomposing the error vector as $r_t = \mathbf{P}_1 r_t + (1 - \mathbf{P}_1) r_t$, this implies that $\text{vec}(V_t) = V_t^\top = \frac{1}{1-\gamma} \mathbf{P}_1 r_t + g_\gamma((1 - \mathbf{P}_1) r_t) \in \mathbb{R}^m$, where $\mathbf{P}_1 = \frac{1}{m} \mathbf{1}\mathbf{1}^\top$ is a projection onto the column space of $\mathbf{1} = [1, 1, \ldots, 1]^\top \in \mathbb{R}^m$, and $g_\gamma$ is a function such that for any $q$ in its domain, $\|g_\gamma(q)\| < c$ for all $\gamma \in (0, 1)$ with some constant $c$ in $\gamma$.

In other words, $\text{vec}(V_t)$ in Theorem 2 can be decomposed into the two terms: $\frac{1}{1-\gamma} \mathbf{P}_1 r_t$ (the projection of the error vector onto the column space of $\mathbf{1}$) and $g_\gamma((1 - \mathbf{P}_1) r_t)$ (a function of the projection of the error vector onto the null space space of $\mathbf{1}$). Here, as $\gamma \to 1$, $\|\frac{1}{1-\gamma} \mathbf{P}_1 r_t\| \to \infty$ and $\|g_\gamma((1 - \mathbf{P}_1) r_t)\| < c$. This implies that with $\gamma < 1$ sufficiently large, the first term $\frac{1}{1-\gamma} \mathbf{P}_1 r_t$ dominates the second term $g_\gamma((1 - \mathbf{P}_1) r_t)$.

Since $(\mathbf{P}_1 r_t)^\top \phi(x) = \frac{\bar{\mu}_t}{m} \mathbf{1}^\top \phi(x)$ with $\bar{\mu}_t = \sum_{k=1}^{m} (r_t)_k \in \mathbb{R}$, this implies that with $\gamma < 1$ sufficiently large, the dynamics of deep equilibrium linear models $\frac{d}{dt} f_{\theta_t} = \frac{1}{\delta_t} V_t \phi$ learns a simple shallow function $\frac{\hat{\mu}_T}{m} \mathbf{1}^\top \phi(x)$ first before learning more complicated components of the functions through $g_\gamma((1 - \mathbf{P}_1) r_t)$, where $\hat{\mu}_T = \int_0^T \bar{\mu}_t dt \in \mathbb{R}$. Here, $\frac{\hat{\mu}_T}{m} \mathbf{1}^\top \phi(x)$ is a simple average model that averages over the features $\frac{1}{m} \mathbf{1}^\top \phi(x)$ and multiplies it by a scaler $\hat{\mu}_t$. Moreover, large $\gamma < 1$ means that we have large *effective depth* or large weighting for deeper layers since we have a shallow model with $\gamma = 0$ and $\gamma$ is a discount factor of the infinite depth.

# C    ADDITIONAL DISCUSSION

**On existence of the limit.** When Bai et al. (2019a) introduced general deep equilibrium models, they hypothesized that the limit $\lim_{l \to \infty} z^{(l)}$ exists for several choices of $h$, and provided numerical results to support this hypothesis. In general deep equilibrium models, depending on the values of model parameters, the limit is not ensured to exists. For example, when $h$ increases the norm of the output at every layer, then it is easy to see that the sequence diverge or explode. This is also true when we set $h$ to be that of deep equilibrium models without the nonlinearity $\sigma$ (or equivalently redefining $\sigma$ to be the identity function): if the operator norms on $A$ are not bounded by one, then the sequence can diverge in general. In other words, in general, some trajectory of gradient dynamics may potentially violate the assumption of the existence of the limit when learning models. In this view, the class of deep equilibrium linear models is one instance of general deep equilibrium models where the limit is guaranteed to exist for any values of model parameters as stated in Proposition 1.

**On the PL inequality.** With $R$ sufficiently large, the definition of the PL inequality in this paper is simply a rearrangement in the form where $L_0$ is allowed to take matrices as its inputs through

$L_0^{\text{vec}}(\text{vec}(\cdot)) = L_0(\cdot)$, where $\text{vec}(M)$ represents the standard vectorization of a matrix $M$. See (Polyak, 1963; Karimi et al., 2016) for more detailed explanations of the PL inequality.

**On the reditus $R$ for the logistic loss.**   As shown in Section 3.1.3, we can use $R = \infty$ for the square loss and the logistic loss, in order to get a prior guarantee for the global linear convergence in theory. In practice, for the logistic loss, we may want to choose $R$ depending on the different scenarios, because of the following observation. For the logistic loss, we would like to set the radius $R$ to be large so that the trajectory on $B$ is bounded as $\|B_t\|_1 < (1 - \gamma)R$ for all $t \in [0, T]$ and the global minimum value on the constrained domain to decrease: i.e., $L_R^* \to L^*$ as $R \to \infty$. However, unlike in the case of the squared loss, the convergence rate decreases as we increase $R$ in the case of the logistic loss, because $\rho(R)$ decreases as $R$ increases. This does not pose an issue because we can always pick $R < \infty$ so that for any $t > 0$ and $T > 0$, we have $\rho(R) > c_\rho$ for some constant $c_\rho > 0$. Moreover, this tradeoff does not appear for the square loss: i.e., we can set $R = \infty$ for the square loss without decreasing the convergence rate. We can also avoid this tradeoff for the logistic loss by simply setting $R = \infty$ and $\tau > 0$.

**On previous work without implicit linearization.**   In Section 5, we discussed the previous work on deep neural networks with implicit linearization via significant over-parameterization. Kawaguchi & Huang (2019) observed that we can also use the implicit linearization with mild over-parameterization by controlling learning rates to guarantee global convergence and generalization performances at the same time. On the other hand, there is another line of previous work where deep nonlinear neural networks are studied without any (implicit or explicit) linearization and without any strong assumptions; e.g., see the previous work by Shamir (2018); Liang et al. (2018); Nguyen (2019); Kawaguchi & Bengio (2019); Kawaguchi & Kaelbling (2020); Nguyen (2021). Whereas the conclusions of these previous studies without strong assumptions can be directly applicable to practical settings, their conclusions are not as strong as those of previous studies with strong assumptions (e.g., implicit linearization via significant over-parameterization) as expected. The direct practical applicability, however, comes with the benefit of being able to assist the progress of practical methods (Verma et al., 2019; Jagtap et al., 2020b;a).

## D   EXPERIMENTS

The purpose of our experiments is to provide a secondary motivation for our theoretical analyses, instead of claiming the immediate benefits of using deep equilibrium linear models.

### D.1   EXPERIMENTAL SETUP

For data generation and all models, we set $\phi(x) = x$. Therefore, we have $m = m_x$.

**Data.**   To generate datasets, we first drew uniformly at random 200 input images from a standard image dataset — CIFAR-10 (Krizhevsky & Hinton, 2009), CIFAR-100 (Krizhevsky & Hinton, 2009) or Kuzushiji-MNIST (Clanuwat et al., 2019) — as pre input data points $x_i^{\text{pre}} \in \mathbb{R}^{m_x'}$. Out of 200 images, 100 images to be used for training were drawn from a train dataset and the 100 other images to be used for testing were drawn from the corresponding test dataset. Then, the input data pints $x_i \in \mathbb{R}^{m_x}$ with $m_x = 150$ were generated as $x_i = \mathbf{R}x_i^{\text{pre}}$ where each entry of a matrix $\mathbf{R} \in \mathbb{R}^{m_x \times m_x'}$ was set to $\delta/\sqrt{m_x}$ with $\delta \overset{\text{i.i.d.}}{\sim} \mathcal{N}(0, 1)$ and was fixed over the indices $i$. We then generated the targets as $y_i = B^*(\lim_{l \to \infty} z^{(l)}(x_i, A^*)) + \delta_i \in \mathbb{R}$ with $\gamma = 0.8$ where $\delta_i \overset{\text{i.i.d.}}{\sim} \mathcal{N}(0, 1)$. Each entry of the true (unknown) matrices $A^* \in \mathbb{R}^{1 \times m}$ and $B^* \in \mathbb{R}^{m \times m}$ was independently drawn from the standard normal distribution.

**Model.**   For DNNs, we used ReLU activation and $W^{(l)} \in \mathbb{R}^{m \times m}$ for $l = 1, 2 \ldots, H - 1$ ($W^{(H)} \in \mathbb{R}^{1 \times m}$). Each entry of the weight matrices $W^{(l)}$ for DNNs was initialized to $\delta/\sqrt{m}$ where $\delta \overset{\text{i.i.d.}}{\sim} \mathcal{N}(0, 1)$ for all $l = 1, 2, \ldots, H$. Similarly, for deep equilibrium linear models, each entry of $A$ and $B$ was initialized to $\delta/\sqrt{m}$ where $\delta \overset{\text{i.i.d.}}{\sim} \mathcal{N}(0, 1)$. Linear models were initialized to represent the exact same functions as those of initial deep equilibrium linear models: i.e., $W_0 = B_0 U_0^{-1}$. We used $\gamma = 0.8$ for deep equilibrium linear models.

**Training.**   For each dataset, we used stochastic gradient descent (SGD) to train linear models, deep equilibrium linear models, and fully-connected feedforward deep neural networks (DNNs). We used the square loss $\ell(q, y_i) = \|q - y_i\|_2^2$. We fixed the mini-batch size to be 64 and the momentum

coefficient to be 0.8. Under this setting, linear models are known to find a minimum norm solution (with extra elements from initialization) (Gunasekar et al., 2017; Poggio et al., 2017). Similarly, DNNs have been empirically observed to have implicit regularization effects (although the most well studied setting is with the loss functions with exponential tails) (e.g., see discussions in Poggio et al., 2017; Moroshko et al., 2020; Woodworth et al., 2020). In order to minimize the effect of learning rates on our conclusion, we conducted experiments with all the values of learning rates from the choices of $0.01, 0.005, 0.001, 0.0005, 0.0001$ and $0.00005$, and reported the results with both the worst cases and the best cases separately for each model (and each depth $H$ for DNNs).

All experiments were implemented in PyTorch (Paszke et al., 2019).

### D.2 ADDITIONAL EXPERIMENTS

In this subsection, we report additional experimental results.

**Additional datasets.** We repeated the same experiments as those for Figure 1 with four additional datasets – modified MNIST (LeCun et al., 1998), SVHN (Netzer et al., 2011), SEMEION (Srl & Brescia, 1994), and Fashion-MNIST (Xiao et al., 2017). We report the result of this experiment in Figure 4. As can be seen from Figures 4, we confirmed qualitatively the same observations as in Figure 1: i.e., all models preformed approximately the same at initial points, but deep equilibrium linear models outperformed both linear models and nonlinear DNNs in test errors after training.

**DNNs without bias terms.** In Figures 1–4, the results of DNNs are reported with bias terms. To consider the effect of discarding bias term, we also repeated the same experiments with DNNs without bias term and reported the results in Figure 5. As can be seen from Figures 5, we confirmed qualitatively the same observations: i.e., deep equilibrium linear models outperformed nonlinear DNNs in test errors.

**DNNs with deeper networks.** To consider the effect of deeper networks, we also repeated the same experiments with deeper DNNs with depth $H = 10, 100$ and $200$, and we reported the results in Figures 6–7. As can be seen from Figures 6–7, we again confirmed qualitatively the same observations: i.e., deep equilibrium linear models outperformed nonlinear DNNs in test errors, although DNNs can reduce training errors faster than deep equilibrium linear models. We experienced gradient explosion and gradient vanishing for DNNs with depth $H = 100$ and $H = 200$.

**Larger datasets.** In Figures 1, we used only 200 data points so that we can observe the effect of inductive bias and overfitting phenomena under a small number of data points. If we use a large number of data points, it is expected that the benefit of the inductive bias with deep equilibrium linear models tends to become less noticeable because using a large number of data points can reduce the degree of overfitting for all models, including linear models and DNNs. However, we repeated the same experiments with all data points of each datasets: for example, we use 60000 training data points and 10000 test data points for MNIST. Figure 8 reports the results where the values are shown with the best learning rates for each model from the set of learning rates $S_{\text{LR}} = \{0.01, 0.005, 0.001, 0.0005, 0.0001, 0.00005\}$ (in terms of the final test errors at epoch = 100). As can be seen in the figure, deep equilibrium linear models outperformed both linear models and nonlinear DNNs in test errors.

**Logistic loss and theoretical bounds.** In Corollary 2, we can set $\lambda_T = \frac{1}{m(1+\gamma)^2}$ to get a guarantee for the global linear convergence rate for any initialization in theory. However, in practice, this is a pessimistic convergence rate and we may want to choose $\lambda_T$ depending on initializations. To demonstrate this, Figure 3 reports the numerical training trajectory along with theoretical upper bounds with initialization-independent $\lambda_T = \frac{1}{m(1+\gamma)^2}$ and initialization-dependent $\lambda_T = \inf_{t \in [0,T]} \lambda_{\min}(D_t)$. As can be seen in Figure 3, the theoretical upper bound with initialization-dependent $\lambda_T$ demonstrates a faster convergence rate.

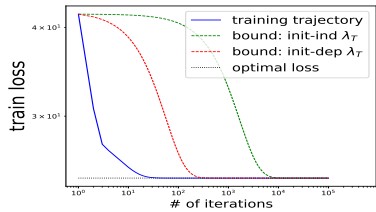

Figure 3: Logistic loss and theoretical bounds with initialization-independent $\lambda_T$ and initialization-dependent $\lambda_T$.

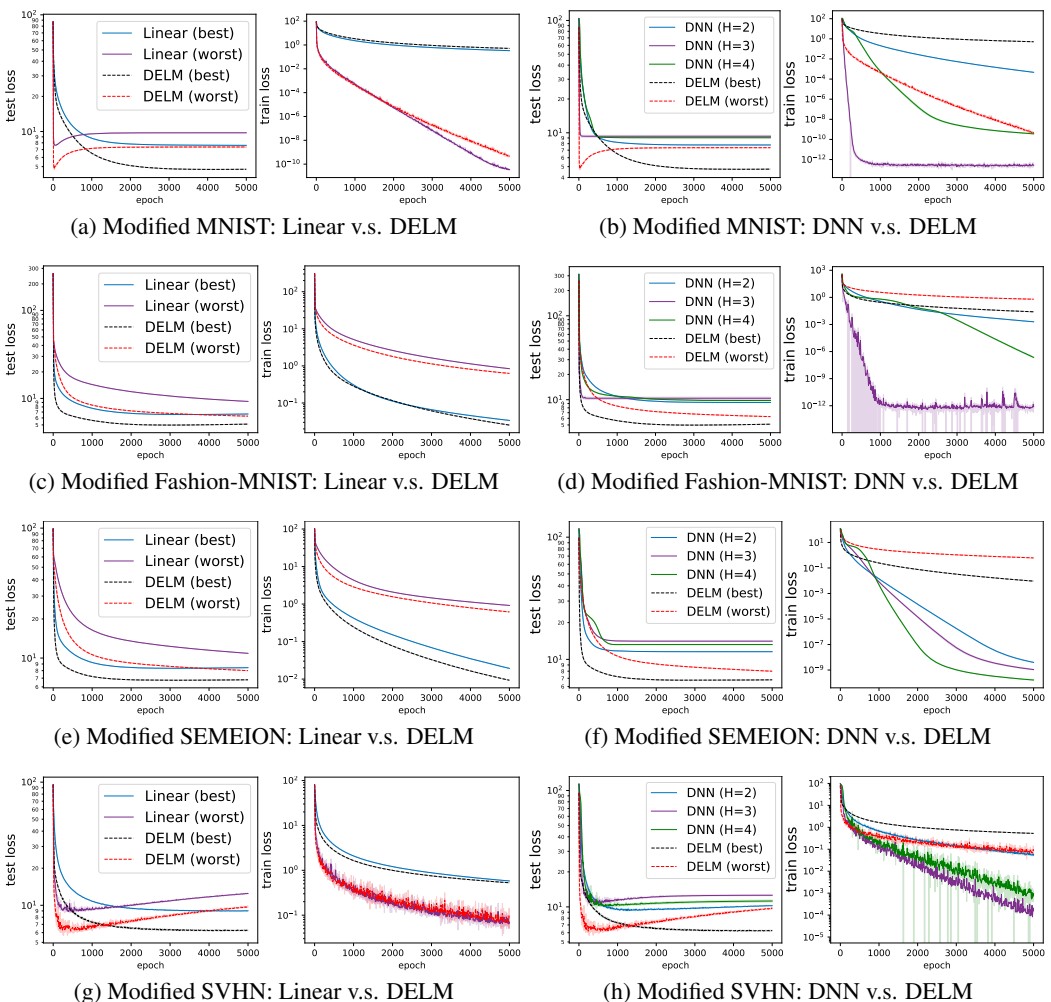

Figure 4: Test and train losses (in log scales) versus the number of epochs for linear models, deep equilibrium linear models (DELMs), and deep neural networks with ReLU (DNNs). The plotted lines indicate the mean values over five random trials whereas the shaded regions represent error bars with one standard deviations for both test and train losses. The plots for linear models and DELMs are shown with the best and worst learning rates (in terms of the final test errors at epoch = 5000). The plots for DNNs are shown with the best learning rates for each depth $H = 2, 3$, and $4$ (in terms of the final test errors at epoch = 5000).

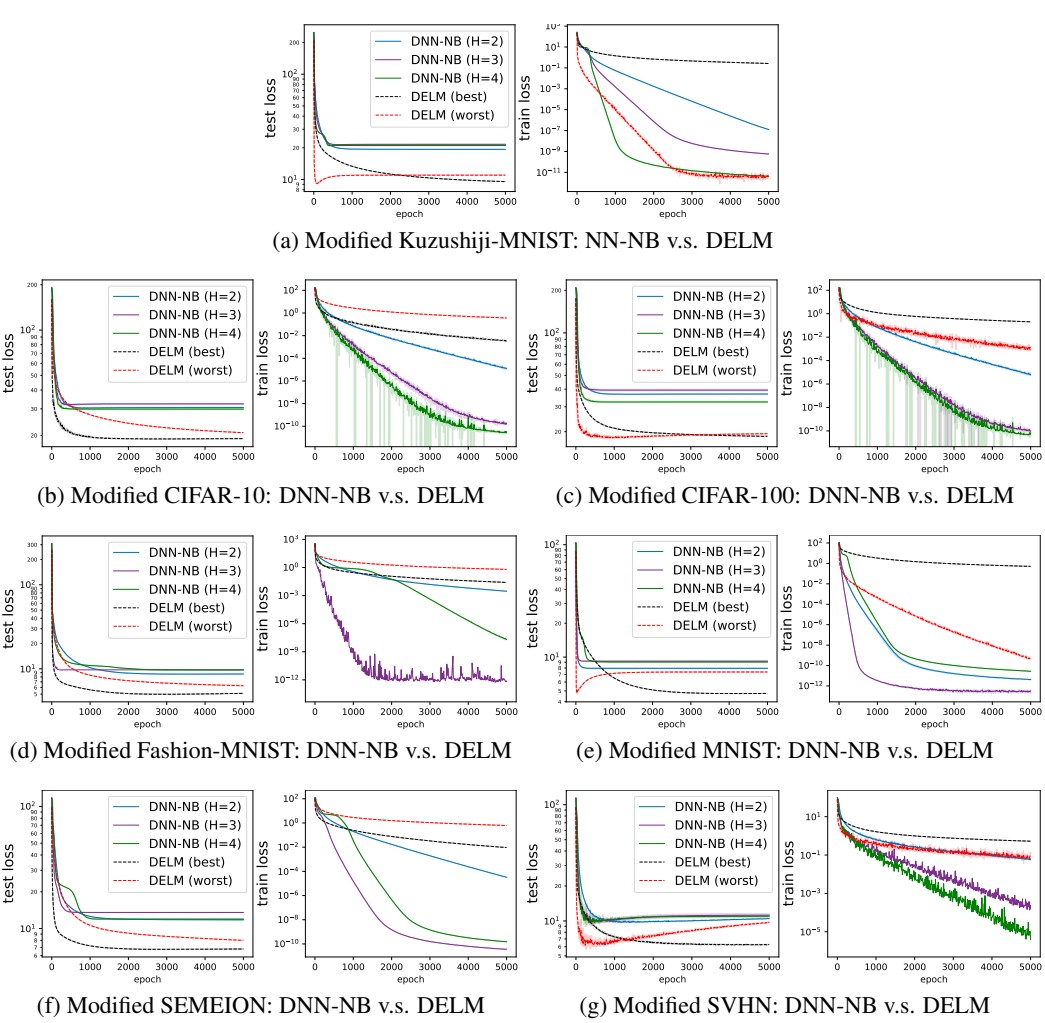

Figure 5: Test and train losses (in log scales) versus the number of epochs for deep equilibrium linear model (DELM) and deep neural network with no bias term (DNN-NB). The plotted lines indicate the mean values over five random trials whereas the shaded regions represent error bars with one standard deviations for both test and train losses. The plots for DELM are shown with the best and worst learning rates (in terms of the final test errors at epoch = 5000). The plots for DNN-NB are shown with the best learning rates for each depth $H = 2, 3$, and $4$ (in terms of the final test errors at epoch = 5000).

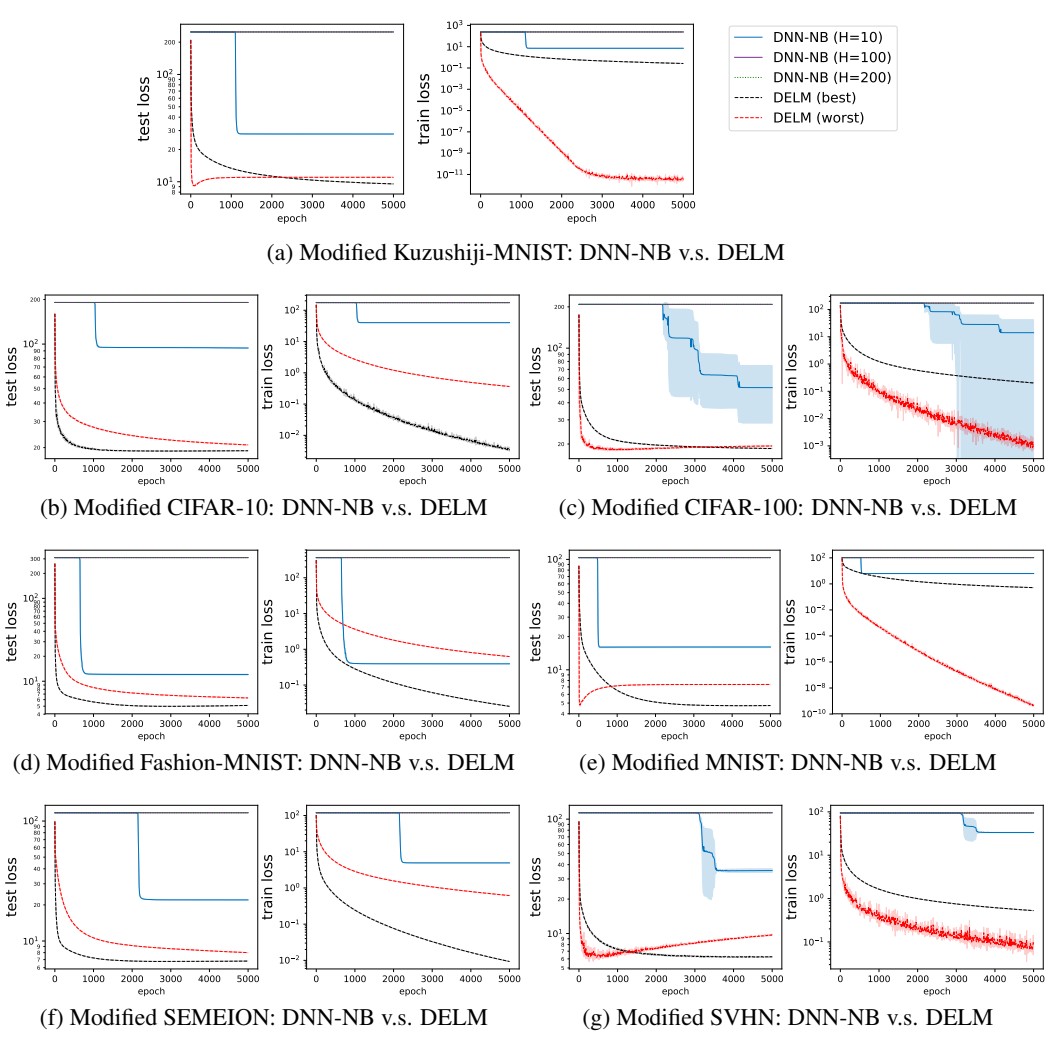

Figure 6: Test and train losses (in log scales) versus the number of epochs for deep equilibrium linear model (DELM) and *deeper* neural network with no bias term (DNN-NB). The legend is the same for all subplots and shown in subplot (a). The plotted lines indicate the mean values over five random trials whereas the shaded regions represent error bars with one standard deviations for both test and train losses. The plots for DELM are shown with the best and worst learning rates (in terms of the final test errors at epoch = 5000). The plots for DNN-NB are shown with the best learning rates for each depth $H = 10, 100$, and $200$ (in terms of the final test errors at epoch = 5000). The values for DNNs with depth $H = 100$ and $200$ coincide in the plots.

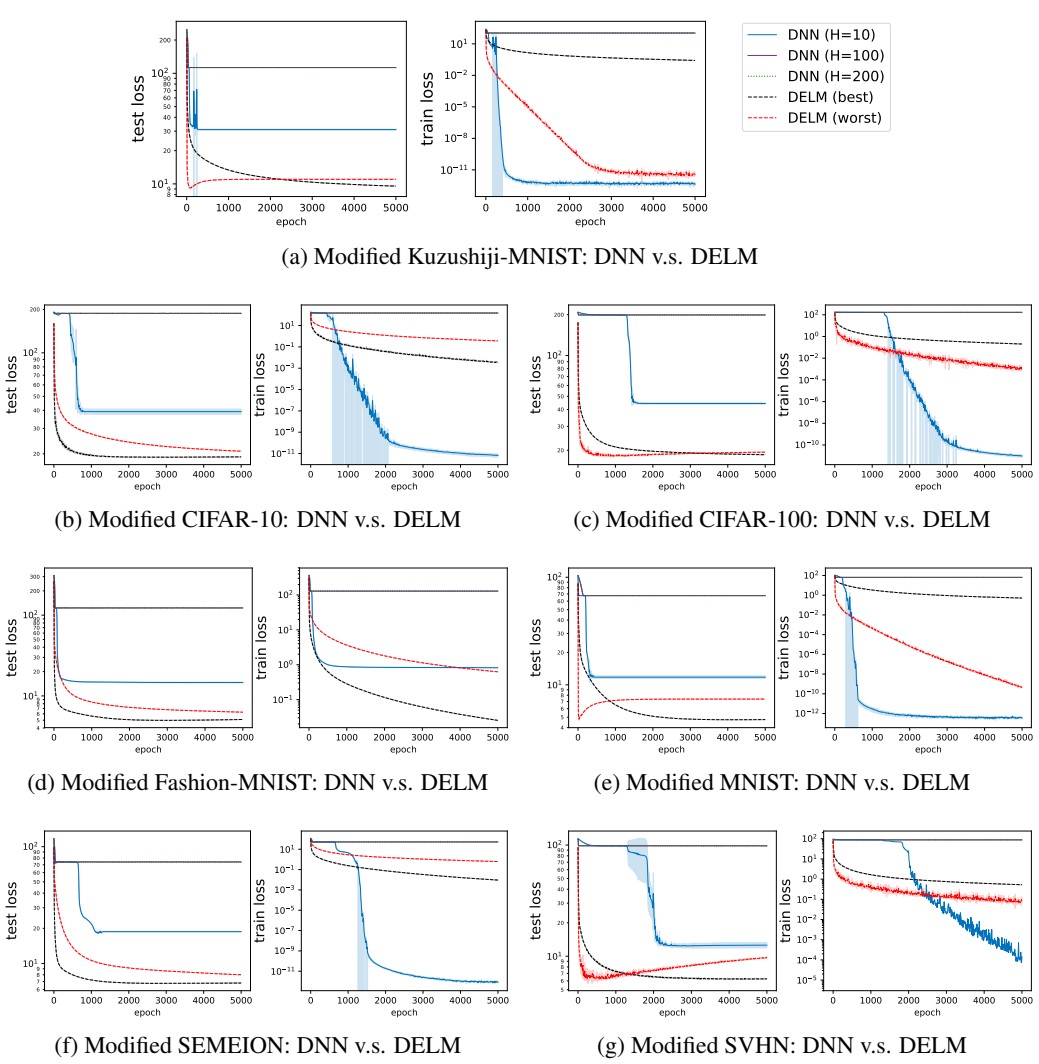

Figure 7: Test and train losses (in log scales) versus the number of epochs for deep equilibrium linear model (DELM) and *deeper* neural networks with ReLU (DNNs). The legend is the same for all subplots and shown in subplot (a). The plotted lines indicate the mean values over five random trials whereas the shaded regions represent error bars with one standard deviations for both test and train losses. The plots for DELM are shown with the best and worst learning rates (in terms of the final test errors at epoch = 5000). The plots for DNNs are shown with the best learning rates for each depth $H = 10, 100$, and $200$ (in terms of the final test errors at epoch = 5000). The values for DNNs with depth $H = 100$ and $200$ coincide in the plots.

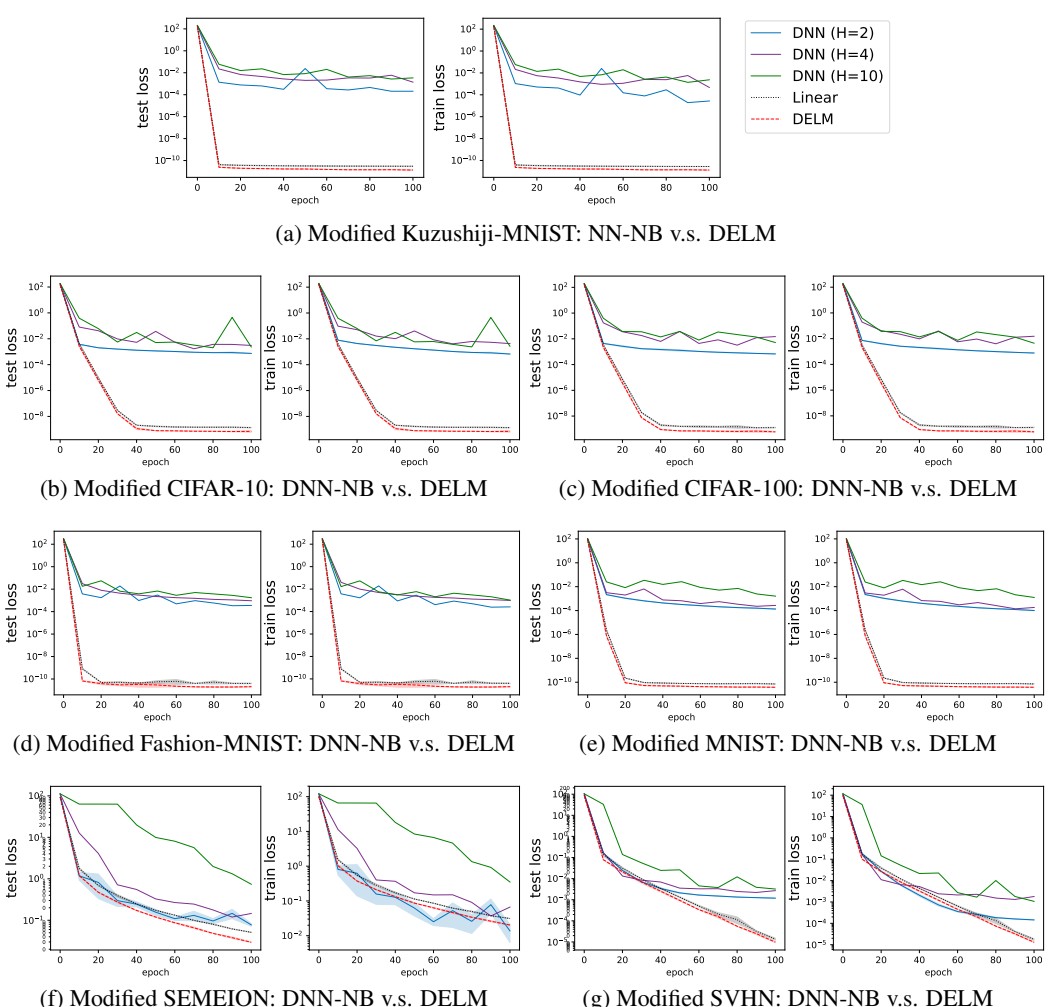

Figure 8: Test and train losses (in log scales) versus the number of epochs for linear models, deep equilibrium linear models (DELMs), and deep neural networks with ReLU (DNNs). The legend is the same for all subplots and shown in subplot (a). The plotted lines indicate the mean values over three random trials whereas the shaded regions represent error bars with one standard deviations.

