# OpenReview forum: "On the Theory of Implicit Deep Learning: Global Convergence with Implicit Layers"
_ICLR.cc/2021/Conference — ICLR 2021 Spotlight_

### Official Review · AnonReviewer1 · 2020-10-23
**R1: Important theoretical study of deep implicit models, though with limited scope**

**Rating:** 7
**Confidence:** 3

**Review:**

> Summary: This work focuses on the study of (global) convergence and gradient dynamics of a recently proposed family of models, the deep equilibrium  (linear) models (DELM) under common classes of loss functions. Exploiting the Neumann series convergence and the PL inequality analysis, the authors proved convergence to global optima of DELM without prior assumption on the width m of the model (relative to the number of data n).

Here is my general opinion:

While deep linear models in general (as the authors acknowledged) has been widely studied, and despite the extreme simplicity of the DELM when compared to the original DEQ model that Bai et al. [1] studied, I found this work *interesting and important* in establishing a solid foundation for the theoretical study of this class of implicit-depth models. The authors managed to demonstrate that the gradient dynamics and convergence assumptions of DELM is __indeed__ different from typical "stacked" deep linear models that prior approaches study, such as deep linear ResNet. And throughout the arguments of the paper and the proof in the appendix, I can tell how the "equilibrium" property of DELM is making the story different, and think this paper sets a good starting point for future similar in this direction for general implicit-depth models. But still, the paper has a limited scope in terms of the structure it studies.

Pros:
1. One of the first theoretical works on the gradient dynamics and convergence properties of the deep equilibrium models [1] (and implicit models in general [2,3]), which are quite different from conventional deep networks.
2. Clear notation and theoretical insights, with proof relatively easy to follow. The proof seems overall correct (there are some that I didn't check closely though).
3. Clear discussion of the relation, including (and especially) the differences of the prior analysis on deep linear neural networks.

Cons:
1. The very definition of DELM, which the author provided a particular formulation of, is of limited scope (see my comment below that expands on this point).
2. The empirical studies to validate the conclusion of the theoretical results could be strengthened.

-------------------------------

I have some comments/questions for the authors, detailed below:

1. The major limitation that I found while reading the discussion and the proof of this paper is that while the authors claim to study deep equilibrium **linear** models, the insights mostly only apply to the models converging with Neumann series guarantee, and can be written in the form $BU^{-1} \phi(x)$. I understand the motivation for fixating a provably convergent equilibrium model formulation. But:

    a) A provably convergent deep equilibrium linear model doesn't need to be Neumann (for its Jacobian) for the implicit function theorem to work. In the simplest case, the fixed point of a function $h(x)$ on 2D can have a local derivative with absolute value > 1. This certainly implies that repeatedly unrolling the function $h(x)$ may not converge, and yet **there still is** a unique fixed point and one can reliably solve for it (see [4]). However, without the nice Neumann series form, which allows one to write $(I-\gamma \sigma(A))^{-1}$ as a closed-form representation for the "infinite-depth" network forward pass, I don't think the theorems will hold directly. Typically, the $(I-J)^{-1}$ term should only appear in the implicit function theorem, which is used for the backward pass. I expect the authors to clarify this further.

    b) The very design of $\sigma(A)$ in the model the authors study is a bit bizarre to me. Why applying a "softmax" on the weight? Is it just to ensure that proposition 1 holds (i.e., that you have a handy, provably-convergent linear model)? The authors stressed a few times that the $h(\cdot)$ function is thus "non-linear" w.r.t. $\sigma(A)$, but I fail to directly see why it matters so much as the model is still linear w.r.t. the input (it's really a one-linear layer; though the inverse from Neumann does make a difference on A), and in terms of the gradient dynamics, the major difference this makes will merely be $\frac{\partial \sigma(A)}{\partial A}$. I might have missed something here and would appreciate if the authors can clarify.

2. I didn't quite get the specific point the authors were trying to make in Section 3.2 in terms of the implicit bias. Could you expand on that?

3. Overall, I feel that the empirical support of the theoretical findings can be stronger, for instance, by inspecting different initialization $(A_0, B_0)$, or validating the radius discussion at the end of Section 3.1 for the logistic loss. Like in Zou et al. [5], some synthetic data could probably work just fine. What is the reason for only using 200 images from the MNIST/CIFAR datasets? Is it to keep the size of $\Phi$ small (but I didn't see the authors report anything about it in Section 2.2). And since the primary purpose of Sec. 2.2. is to "discuss whether the model would also make sense in practice" (which I take to mean that you only want to compare the test accuracies coming out of these models), wouldn't a 200-sample version of MNIST/CIFAR too small to draw a robust conclusion on this?

4. One that I think could be useful for further thought is the convergence property not just for GD, but also SGD, like Zou et al. provided in [5] (up to a probability).

-------------------------------

Minor things:

  - i) Page 3: "the outputs of the deep equilibrium linear models $f_\theta(x)= \cdots$ are nonlinear and non-multilinear in the optimization variables $(A,B)$." Non-linear in even $B$?
  - ii) Page 14: $V_{q*}$ --> $B_{q*}$
  - iii) Page 16: $\nabla_F L(A,B)$ --> $\nabla_A L(A,B)$

-------------------------------

- [1] https://arxiv.org/abs/1909.01377
- [2] https://arxiv.org/abs/1908.06315
- [3] https://arxiv.org/abs/2009.06211
- [4] https://arxiv.org/abs/2006.08591
- [5] https://arxiv.org/abs/2003.01094

---

> ### Author Response · Authors · 2020-11-22
> **Response to Reviewer 1**
>
> Thank you for the review and remarking that this work is “interesting and important in establishing a solid foundation for the theoretical study of this class of implicit-depth models”. We extensively revised the paper according to all of your comments by adding an additional page to the main text (as allowed by ICLR 2021 during this phase) and adding more results in the appendix. These revisions increased the length of the paper from 30 pages to 37 pages. We believe that this improved the paper a lot, thanks to your comments. Below we respond to your comments and questions.
>
>
> == Con 1 and Question 1 ==
>
> We revised the paper by adding the discussion of the alternative proofs using the implicit function theorem to avoid relying on the convergence of Neumann series in Appendix A. In our default proofs, we utilize the Neumann series when deriving the formula of the gradients with respect to A. However, we can instead use the implicit function theorem to derive the formula of the gradients with respect to A as shown in the added alternative proofs.
>
> On the other hand, we agree with you that the results of this paper does not *directly* apply to some of more general cases, although the proof idea and many proof steps would apply. However, limiting the scope in this way is a crucial key step in the progress of the theory in many fields, including the theory of deep learning (for example, see [1]-[9]), theoretical physics, and theoretical economics. Also, please note that the most previous papers in the related literature focus on the square loss where we have closed-form solutions. For the logistic loss, we don’t have a closed-form solution in general. Therefore, there is a gap between them, similarly to the gap mentioned in this question. Despite this gap, the analysis and the proof idea for the square loss turned out to be useful for the logistic loss. Similarly, we believe that our analysis and proof idea will be useful for some of the more general cases that are not directly covered in this paper.
>
> For Question 1 (b), this nonlinearity σ is one of the natural choices because of the following observation. We can view σ(A) as a transition matrix in a stochastic process of $z^{(l)} = h(z^{(l-1)}; x, \theta)$, where σ(A) represents a conditional probability as σ(A)_{ij} = P(i | j). A natural way to encode this information is to set σ to be a softmax function. That is, the underlying reason of using softmax here is the same as the reason why we typically use softmax at the last layer of neural networks for multi-class classification. As the reviewer noted, this also ensures that Proposition 1 holds. But, we can prove Proposition 1 for different choices of the nonlinearity, and this is not the main reason as mentioned above. For the comment on the linearity on the input, we can always make it nonlinear in the input by choosing the nonlinear features $\phi$, and also we can use $\phi$ that approximates NTK of deep nonlinear networks, to make it related to deep nonlinear networks. In the analysis of gradient dynamics, the major challenge is the nonlinearity in terms of in the parameters that we are optimizing over. Indeed, in the proofs, we can see that the nonlinearity σ creates the interactions among different neurons/coordinates, which cannot be disentangled and cannot be dealt with by proofs in the previous papers. Such interactions among different coordinates are often the main challenges in the analysis of gradient dynamics of deep networks, in both linear networks (e.g., [1] assumes a particular initialization to avoid this interactions) and nonlinear networks (e.g., previous papers assume implicit linearization with the NTK regime to avoid this interactions: i.e., in the NTK regime, there is no such interaction as NTK stays nearly the same during training). As far as we know, our paper is the first paper to directly analyze and deal with this type of challenging interactions without assuming sparsity, linear separability, or Gaussian distribution.
>
> [1] Exact solutions to the nonlinear dynamics of learning in deep linear neural networks. Saxe et al. ICLR 2014.
>
> [2] Deep learning without poor local minima. Kawaguchi. NeurIPS 2016.
>
> [3] Identity matters in deep learning. Hardt and Ma. ICLR 2016.
>
> [4] On the optimization of deep networks: Implicit acceleration by overparameterization. Arora et al. ICML 2018.
>
> [5] Deep linear networks with arbitrary loss: All local minima are global. Laurent and Brecht. ICML 2018.
>
> [6] On the optimization of deep networks: Implicit acceleration by overparameterization. Arora et al. ICML 2018.
>
> [7] A convergence analysis of gradient descent for deep linear neural networks. Arora et al. ICLR 2019.
>
> [8] Width provably matters in optimization for deep linear neural networks. Du and Hu. ICML 2019.
>
> [9] On the global convergence of training deep linear ResNets. Zou et al. ICLR 2020.
>
> ====
>
> [Continued in the next post...]

---

> > ### Author Response · Authors · 2020-11-22
> > **Response to Reviewer 1: continued**
> >
> > == Con 2 and Question 3 ==
> >
> > Thank you for your comments on the empirical study to validate the conclusion of the theoretical results. We believe that this comment improved our paper a lot. Namely, we revised the paper according to this comment as follows:
> >
> > (1) We added new experimental results for different initializations in Section 4. The new results verify our theory on the global convergence of deep equilibrium linear models with any initialization and all random trials. Following your comment, we used the dataset from the previous paper (Zou et al., 2020b).
> >
> > (2) We added a new theory result for the logistic loss, a new version of Corollary 2, and revised the paragraphs for the discussion on the radius for the logistic loss. With the new version of Corollary 2, we can also set $R=\infty$ to remove the notion of the radius $R$ from the statement of the global convergence for the logistic loss. Therefore, for both the square loss and the logistic loss, there is no need for the discussion on the radius. On the other hand, we still have the discussion about different types of bounds implied by our theory results, similarly to the discussion on the radius in the original submission. Accordingly, we revised the paper as mentioned below in (3).
> >
> > (3) We added new experimental results and discussions for validating our theory on various global convergence rates with different $\lambda_T$ values that depend on initializations. The results are reported in Section 4 for the square loss, and in Appendix D.2 for the logistic loss.
> >
> > (4) We added new experimental results with all data points (instead of 200-sample version) in Appendix D.2, and confirmed that deep equilibrium linear models still outperformed other models.
> >
> > We will now answer the following question: “What is the reason for only using 200 images from the MNIST/CIFAR datasets?”. The reason is that we can observe the effect of inductive bias and overfitting phenomena under a small number of data points better than under a large number of data points. The regime of the few data points is an important and interesting regime as well: i.e., it is not an inferior setting when compared to a setting with many data points. Each has its own merit. If we use a large number of data points, it is expected that the benefit of the inductive bias tends to become less noticeable because using a large number of data points can reduce the degree of overfitting for all models, including linear models and DNNs. Therefore, it is sensible to study overfitting phenomena under the setting of a small number of data points as well. As we increase the number of samples, the gap and the effect of the inductive bias simply reduce since having more samples helps all models to mitigate overfitting.
> >
> >
> > The answer to the following question is No: “Is it to keep the size of Φ small (but I didn't see the authors report anything about it in Section 2.2).” The size of \phi(x) does not increase as we increase the number of data points, and thus the computational time per iteration does not increase as we increase the number of data points. Increasing the size of Φ in the dimension of n does not increase the computational time per iteration. Indeed, our theory suggests that the worst case is when n = d, as that is when the minimum singular value of Φ can be likely small. If n or d are much larger than each other, there are many components that are summed up for Φ^\top Φ or Φ Φ^\top, and the minimum singular value of Φ tends to increase and thus the convergence rate tends to get better.
> >
> >
> > For the robustness of the conclusion with 200-sample versions, we explained above that we added the new results with all samples and that the 200-sample versions themselves make sense to be studied for drawing the conclusion as that is in the regime of own interest for inductive bias. Please see above for more details.
> >
> >
> > == Question 2 ==
> >
> > According to the question, we extensively revised Section 3.2 by adding clarifications, proving a new clearer version of Theorem 2 (with the revised proof in Appendix A.2), and adding a new result on implicit bias in Appendix B. The new version of Theorem 2 more clearly illustrates potential benefits of deep equilibrium linear models in two aspects: when compared to shallow models, it can sometimes accelerate optimization via the effect of the implicit trust region method (but not necessarily as the trust region method does not necessarily accelerate optimization) and induces novel implicit bias for generalization via the non-standard implicit trust region. Moreover, in the newly added appendix, Appendix B, we show that this new version of the second theorem suggests an implicit bias towards a simple function as a result of infinite depth.
> >
> >
> >
> > == Question 4 ==
> >
> > This is an interesting future work. Thank you for the comment! We revised the paper by adding this as a future work.
> >
> > ====
> >
> > [Continued in the next post...]

---

> > > ### Author Response · Authors · 2020-11-22
> > > **Response to Reviewer 1: continued**
> > >
> > > == Minor things ==
> > >
> > > i) it is nonlinear in (A, B) and also *not* multi-linear in (A, B) as stated. For example, f(x,y) = xy is nonlinear in (x, y) but multi-linear in (x, y). But, to be clearer, we revised the paper as: “… nonlinear and non-multilinear in the optimization variable A”, without mentioning B.
> > >
> > > ii) & ii) We fixed these typos. Thank you!
> > >
> > > Thank you very much for all of your comments!

---

> ### Comment · AnonReviewer1 · 2020-11-22
> **Update after the response from the authors**
>
> I would like to thank the authors for updating the paper with the various new discussions on the points that I raised in the review. The experimental results, though (understandably) still on a small scale like many other theory papers, have better corroborated the claim of the authors. In addition, the alternative proof by IFT is a lot more straightforward than I initially expected (thanks to the linearity of the structure assumed in the DELM, actually) :-)
>
> I have updated my score to 7 accordingly. This is important & interesting work!

---

> > ### Author Response · Authors · 2020-11-22
> > **Thank you**
> >
> > Thank you for the positive feedback!

---

### Official Review · AnonReviewer3 · 2020-10-28
**a clear analysis of deep equilibrium models**

**Rating:** 7
**Confidence:** 3

**Review:**

This submission studies the dynamics and convergence properties of "deep equilibrium models", which are parametric fixed-point iterations corresponding to the infinite depth limit of "weight-tied" neural networks. As the authors point out, these networks differ from deep linear networks and networks in the NTK scaling in that the optimization remains nonlinear w/r/t the parameters. The authors prove two results: first, they establish linear convergence to the global minimum under the relatively strict assumption of a "local" PL-inequality; secondly, they show that the dynamics of the deep equilibrium models differs from gradient descent dynamics and, in fact, is related to a trust region Newton method.

The first theorem is nice and well-presented, but I think the issue of the radius over which the PL-inequality holds could be better discussed. I could not tell whether or not the convergence depends on starting within the locally smooth and quadratically bounded region of the loss.

The second theorem, regarding the nature of the dynamics, lacks a clear interpretation. Basically, all that is said is that the dynamics is distinct from what would be seen in a linear model, and it's not clear that this dynamics has anything to do with implicit regularization, as the authors suggest. I would recommend the authors clarify the discussion of this result.

The experiments are somewhat bizarre and I felt that they were a little misleading about the representative power / potential of these models, but perhaps I did not fully understand the set-up and intent. The authors randomly sample a deep equilibrium model and then use it to represent a conditional probability distribution where the conditioning is with image data. Then, the authors fit various models to this distribution and show that the deep equilibrium models (which is precisely the underlying function representing the distribution) has better performance than other classes of functions. I think the description in this section could be vastly improved, perhaps presenting this more akin to a student-teacher problem.

---

> ### Author Response · Authors · 2020-11-22
> **Response to Reviewer 3**
>
> Thank you for the positive support and the suggestions for further improvements! They allowed us to improve the paper a lot. We revised the paper according to all of your comments as described below.
>
>
> ====
>
> For the suggestion on the clearer discussion on the radius over which the PL-inequality holds, we revised the paper by adding several clarifications, proving a new version of the global convergence result for the logistic loss (Corollary 2), and adding a new experiment section (Section 4). In particular, our convergence results do *not* depend “on starting within the locally smooth and quadratically bounded region of the loss”.
>
> To clarify about the radius, we added the following sentence for the square loss: in Corollary 1, the global linear convergence is established for the square loss without the notion of the radius $R$ as we set $R=\infty$.
>
> To further clarify about the radius, we extensively revised the paragraphs after Corollary 2. In particular, in Corollary 2, we can also set $R=\infty$ to remove the notion of the radius $R$ from the statement of the global convergence for the logistic loss.
>
> Therefore, for both the square loss and the logistic loss, there is no “issue of the radius over which the PL-inequality holds”. We believe that these clarify the meaning of the radius in Theorem 1 with the concrete choices of the common loss functions (square loss and the logistic loss), which is clearer than discussing it abstractly without concrete loss functions.
>
> We further clarify this point in the new experiment section, Section 4, where we demonstrate the global convergence for different initializations. For example, we observe that deep equilibrium linear models converge to the global minimum value with any initialization and all random trials, whereas linear ResNet converges to a suboptimal value with identity initialization. These observations are consistent with our theory where deep equilibrium linear models are guaranteed to converge to the global minimum value without any condition on the initialization.
>
> ====
>
> For the comment on “a clear interpretation” for “the second theorem”, we extensively revised Section 3.2 by adding clarifications, proving a new clearer version of the second theorem (Theorem 2) (with the revised proof in Appendix A.2), and adding a new result on implicit bias in Appendix B. The new version of the second theorem (Theorem 2) more clearly illustrates potential benefits of deep equilibrium linear models in two aspects: when compared to shallow models, it can sometimes accelerate optimization via the effect of the implicit trust region method (but not necessarily as the trust region method does not necessarily accelerate optimization) and induces novel implicit bias for generalization via the non-standard implicit trust region. Moreover, in the newly added appendix, Appendix B, we show that this new version of the second theorem suggests an implicit bias towards a simple function as a result of infinite depth.
>
>
> ====
>
> For the experiments, we revised the paper by adding new experimental results with larger datasets and the dataset from the previous paper (Zou et al., 2020b) in Section 4 and Appendix D.2. The dataset from the previous paper (Zou et al., 2020b) is not generated by deep equilibrium linear models. These new experiments verify our theory with different initializations and with comparisons of the numerical trajectory and our theoretical upper bonds. The purpose of our experiments is to provide a secondary (not primary) motivation for our theoretical analyses and to verify our theory, instead of claiming the immediate benefits of using deep equilibrium linear models. We added this point at the end of Section 2.2. As a theory paper, the purpose of experiments in this paper differs from that of an experiment paper.

---

### Official Review · AnonReviewer4 · 2020-10-28
**Convergence guarantees for linear deep equilibrium models**

**Rating:** 7
**Confidence:** 3

**Review:**

The paper discusses the theory of deep equilibrium models with linear activations. The model weights are softmaxed to ensure that inference converges to a fixed point, a necessary condition for training deep equilibrium models. The paper then analyzes the gradient flow dynamics of such models. The main result is that linear-rate convergence is guaranteed for a class of loss functions, including quadratic and logistic losses, when training with gradient flow. This conclusion is supported by experiments conducted in a teacher-student-like setup, where the labels are generated by a teacher deep equilibrium model, showing that training does converge in practice.

Deep equilibrium models represent a novel way to train neural networks, and not much is known about them yet theoretically. It is important that we understand the dynamics of such models better, and this paper is a good step in that direction.

Suggested improvements:

1. Definition 1 has a typo: On the right-hand side of the inequality there shouldn't be a gradient.

2. The main results are presented clearly, and the paper is generally easy to read. The only exception is section 3.2 on the connection with trust region Newton methods, which I did not understand. I recommend clarifying the intuition behind Theorem 2, as well as the main message of this section.

---

> ### Author Response · Authors · 2020-11-22
> **Response to Reviewer 4**
>
> Thank you for the positive feedback and remarking on the importance of theoretically understanding deep equilibrium models. Also, we highly appreciated the reviewer’s suggestions for further improvements, as they allowed us to improve the paper. Thank you! We revised the paper according to all of your suggestions as described below.
>
>
> ====
>
>
> Suggested improvement 1 (on Definition 1): Thank you for this suggested improvement! We fixed this typo and double-checked that the proofs are consistent with the correct definition without this typo.
>
>
> ====
>
>
> Suggested improvement 2 (on the clarity of section 3.2): We extensively revised Section 3.2 by proving a new clearer version of Theorem 2 (with the revised proof in Appendix A.2) and adding clear intuitions behind this new version of Theorem 2. The new version of Theorem 2 more clearly illustrates potential benefits of deep equilibrium linear models in two aspects: when compared to shallow models, it can sometimes accelerate optimization via the effect of the implicit trust region method (but not necessarily as the trust region method does not necessarily accelerate optimization) and induces novel implicit bias for generalization. Moreover, in Appendix B, we additionally show that Theorem 2 suggests an implicit bias towards a simple function as a result of infinite depth.

---

### Official Review · AnonReviewer2 · 2020-10-28
**Nice framework but not addressing the central question**

**Rating:** 8
**Confidence:** 3

**Review:**

**Overview**

This paper purports to study training deep equilibrium models by studying the optimization dynamics of deep *linear* equilibrium models. The original deep equilibrium model is formalized as follows: Given a training dataset $(x_i,y_i)$ for i = 1,...,n where $x_i \in \mathcal{X}\subseteq \mathbb{R}^{m_x}$ and $y_i\in \mathcal{Y}\subseteq \mathbb{R}^{m_y}$ are the $i$-th input and output, respectively. The goal is to learn a predictor from a family $\mathcal{H} = \\{\{ f_\theta : \mathbb{R}^{m_x} \rightarrow \mathbb{R}^{m_y} | \theta\in\Theta\}\\}$. Then, instead of trying to map $x$ to $y$ using finite amount of layers, deep equilibrium models assume infinite number of layers, and the output $z^*$ of the last hidden layer is defined by
\begin{equation}
    z^* = \lim_{l\rightarrow \infty} z^{(l)} = \lim_{l\rightarrow \infty} h(z^{(l-1)};x,\theta) = h(z^*;x,\theta)
\end{equation}
where $h$ is some continuous function of choice.

In particular with deep equilibrium **linear** models, $h$ is constrained as follows:
\begin{equation}
    h(z^{(l-1)};x,\theta) = \gamma\sigma(A)z^{(l-1)}+\phi(x)
\end{equation}
where $\phi(x)$ is a feature map of $x$ and transforms $x\in\mathbb{R}^{m_x}$ into $\phi(x)\in\mathbb{R}^m$. $\theta = (A,B)$ are two trainable matrices, where $A\in\mathbb{R}^{m\times m}$ is for computing each hidden output and $B\in\mathbb{R}^{m_y\times m}$ is for computing the final output of the network. $\gamma \in (0,1)$ is some positive real number, and $\sigma$ is a nonlinear function to ensure the existence of the fixed point $z^*$. This model is linear in z. The objective function of this deep equilibrium linear model can be written as follows:
\begin{equation}
    L(A,B) = \sum_{i=1}^n \ell\left( B\left( \lim_{l\rightarrow\infty}z^{(l)}(x_i,A)\right),y_i\right)
\end{equation}
where $\ell$ is some choice of loss function.

Then this paper provides some motivations behind studying the dynamics of these deep equilibrium linear models by presenting some interesting comparisons between deep equilibrium linear models and "normal" linear models and additional, less interesting comparisons with fully-connected feed-forward deep neural networks (FNN), using standard image datasets CIFAR-10, CIFAR-100 and Kuzushiji-MNIST. In their tests, the deep equilibrium linear models outperformed both linear models and FNNs.

The main results from this paper is uncovering the dynamics behind these deep equilibrium linear models. This paper provided a sequence of proofs that shows linear convergence of these models step by step, under the assumption that the loss functions $\ell$ are differentiable, which is satisfied by some standard loss functions, such as square loss, losgistic loss, and smoothed hinge loss $\ell(f_\theta(x),y) = (\max\{0,1-f_\theta(x)y\})^k$ with $k\geq 2$. Also, despite of non-convexity of the loss functions, it is shown in this paper that a global minimum $L^*$ always exists and under their assumptions, the deep equilibrium linear models will always converge to the global minimum linearly.

**Strengths**

The main strength of this paper is that it brought  some interesting insights into deep equilibrium  models, and they showed their results rigorously. The definitions and propositions are clear enough for readers with some analysis and machine learning background to fully understand.

Since the dynamics of deep learning models is an open field of research and isn't discovered fully, this paper will definitely contribute to the deep learning field in understanding the dynamics and convergence theory of these deep equilibrium linear models, and potentially benefiting researches on understanding more general deep learning models.

**Weakness**

Although this paper brought some nice ideas, some of the methodologies are not quite convincing. For example, in the experiments shown in this paper, they  compare performance of deep equilibrium linear models with linear models and deep neural networks. Especially in the comparison with the deep networks, it doesn't seem that the networks are deep enough for the comparison to be compelling. Also in the same experiments, they assumed the true data distribution is approximately given by a deep equilibrium linear model and generated data according to this model -- i.e. the data is only semi-"real". It would be better to show that the deep equilibrium linear model outperforms other models under a more general setting.

Throughout the entire paper, it's unclear what are the contributions. To be specific, in the first two sections, it's unclear whether they want to show the trainability of the deep equilibrium linear models or learn the dynamics of these models. Also, some of the interesting aspects of this paper were missing some details. For example, one would be interested in seeing why exactly can the deep equilibrium linear models outperform linear models, since they are both linear and the only difference is how they are trained. In the last two sections of this paper, they also brought up *implicit bias*, which would be another interesting topic to dive into. It would be great seeing more comparisons on those aspects.

---

> ### Author Response · Authors · 2020-11-22
> **Response to Reviewer 2**
>
> Thank you for the feedback and positive comments that our paper provides “interesting insights” and “will definitely contribute to the deep learning field in understanding the dynamics and convergence theory of these deep equilibrium linear models”. We revised the paper according to all the comments by adding an additional page to the main text (as allowed by ICLR 2021 during this phase) and adding more results in the appendix. These revisions increased the length of the paper from 30 pages to 37 pages. We believe that this improved the paper a lot, thanks to your comments. Below we respond to your comments and questions by the order in which they appear.
>
>
> ====
>
>
> For the comparison with the deep networks in experiments, we added new results with deeper networks with depth H = 10, 100, 200 in Appendix D.2. With these deeper networks, we confirmed qualitatively the same observations: deep equilibrium linear models outperformed nonlinear DNNs in test errors.
>
> For experiments in general, we also added new experimental results with larger datasets and the dataset from the previous paper (Zou et al., 2020b) in Section 4 and Appendix D.2. The dataset from the previous paper (Zou et al., 2020b) is not generated by deep equilibrium linear models. These new experiments verify our theory with different initializations and with comparisons of the numerical trajectory and our theoretical upper bonds.
>
> We believe that these new results address the comments for experiments. The purpose of our experiments is to provide a secondary (not primary) motivation for our theoretical analyses and to verify our theory, instead of claiming the immediate benefits of using deep equilibrium linear models. We also added this point at the end of Section 2.2. As a theory paper, the purpose of experiments in the present paper differs from that of an experiment paper.
>
> As stated in Sections 1 and 5, gradient dynamics of deep networks with some linearized component is of interest primarily because they are viewed as a first step in theoretical analysis of deep networks. Hence, there are extensive past works on linearized deep networks as cited in Section 5, without the types of experiments that are usually seen in experiment papers.
>
> ====
>
> Thank you for the following comment: “in the first two sections, it's unclear whether they want to show the trainability of the deep equilibrium linear models or learn the dynamics of these models.”. We revised the end of the first section accordingly to clarify this point. That is, we mathematically prove convergence of gradient dynamics to global minima and the exact relationship between the gradient dynamics of deep equilibrium linear models and that of the adaptive trust region method.
>
> ====
>
> [Continued in the next post...]

---

> > ### Author Response · Authors · 2020-11-22
> > **Response to Reviewer 2: continued**
> >
> > For the comment on “addressing the central question” and “some of the interesting aspects”, we proved a new theorem (the new version of Theorem 2) in Section 3.2 (with the proof in Appendix A.2) and added a new result on implicit bias in Appendix B. The new theorem in Section 3.2 now clearly shows the relationship between deep equilibrium linear models and linear models in the space of the hypothesis. Namely, the implicit bias is induced by the dynamics of V_t that follows the dynamics of the adaptive trust region method of linear models. This suggests potential benefits of deep equilibrium linear models in two aspects: when compared to shallow models, it can sometimes accelerate optimization via the effect of the implicit trust region method (but not necessarily as the trust region method does not necessarily accelerate optimization) and induces novel implicit bias for generalization via the non-standard implicit trust region. In Appendix B, we show that this new theorem suggests an implicit bias towards a simple function as a result of infinite depth.
> >
> > We believe that these extensive revisions addressed your comment for “addressing the central question” and “some of the interesting aspects”. If you agree, please skip the following. If you disagree, there might be a miscommunication between us about this literature, which we want to clarify below: i.e., although we added new results for implicit bias and generalization according to this comment, optimization and trainability itself is one of the *central* questions in the theory of deep learning, as evidenced by many previous papers on the optimization and trainability: for example, see [1]-[9] below. Even if one is interested in generalization instead, the theory of optimization and trainability is still important because the theory of optimization plays a crucial role for theoretically understanding generalization as well. For example, most generalization bounds are written in the following form: generalization error <= training error + a bound on the difference between the generalization error and training error. Here, the bound on the difference is the subject of the study of many generalization theories. As can be seen, along with such generalization theories, we need optimization theory to understand the training error, in order to understand the generalization error. In another type of generalization theory, one analyzes the generalization error of the global minima, which again requires the optimization theory to ensure the global minima. Also, our analysis of the dynamics itself can benefit to understand generalization as discussed above. Beyond generalization, the theoretical advancement of non-convex optimization itself is of great importance as many real-world problems are non-convex and any theory ensuring global minima can have an impact in different fields. These different types of theories are known to each play an important role (and often benefit each other with the connections such as ones mentioned above), and it would not be sensible to discriminate one literature against another in this sense. In any case, your comments helped us improve the paper a lot. Thank you very much for all of your comments!
> >
> > [1] Exact solutions to the nonlinear dynamics of learning in deep linear neural networks. Saxe et al. ICLR 2014.
> >
> > [2] Deep learning without poor local minima. Kawaguchi. NeurIPS 2016.
> >
> > [3] Identity matters in deep learning. Hardt and Ma. ICLR 2016.
> >
> > [4] On the optimization of deep networks: Implicit acceleration by overparameterization. Arora et al. ICML 2018.
> >
> > [5] Deep linear networks with arbitrary loss: All local minima are global. Laurent and Brecht. ICML 2018.
> >
> > [6] On the optimization of deep networks: Implicit acceleration by overparameterization. Arora et al. ICML 2018.
> >
> > [7] A convergence analysis of gradient descent for deep linear neural networks. Arora et al. ICLR 2019.
> >
> > [8] Width provably matters in optimization for deep linear neural networks. Du and Hu. ICML 2019.
> >
> > [9] On the global convergence of training deep linear ResNets. Zou et al. ICLR 2020.

---

### Decision · Program_Chairs · 2021-01-07
**Final Decision**

**Decision:**

Accept (Spotlight)

**Comment:**

The paper analyzes the gradient flow dynamics of deep equilibrium models with linear activations and establishes linear convergence for quadratic loss and logistic loss; several exciting results and connections, solid contribution, accept!